# Coexisting divergent and convergent plate boundary assemblages indicate plate tectonics in the Neoarchean

Bo Huang [1], Tim E. Johnson [2], Simon A. Wilde [2], Ali Polat [3], Dong Fu [1] & Timothy Kusky [1]

The coexistence of divergent (spreading ridge) and convergent (subduction zone) plate boundaries at which lithosphere is respectively generated and destroyed is the hallmark of plate tectonics. Here, we document temporally- and spatially-associated Neoarchean (2.55–2.51 Ga) rock assemblages with mid-ocean ridge and supra-subduction-zone origins from the Angou Complex, southern North China Craton. These assemblages record seafloor spreading and contemporaneous subduction initiation and mature arc magmatism, respectively, analogous to modern divergent and convergent plate boundary processes. Our results provide direct evidence for lateral plate motions in the late Neoarchean, and arguably the operation of plate tectonics, albeit with warmer than average Phanerozoic subduction geotherms. Further, we surmise that plate tectonic processes played an important role in shaping Earth's surficial environments during the Neoarchean and Paleoproterozoic.

Earth is the only habitable planet known to have voluminous felsic continental crust and to operate in a plate tectonic mode, which promotes interaction between the deep interior and the surface of Earth, fundamentally shaping the environment in which we live. However, when plate tectonics first emerged and how it has evolved thereafter, as the mantle progressively cooled, are topics of longstanding and vigorous debate[1–3]. Opinions on when plate tectonics became the dominant mode of planetary cooling on Earth include the Hadean[4–6], the Eoarchean[7], the Meso- to Neoarchean[2,8,9], the Paleoproterozoic[10], and the Neoproterozoic to early Cambrian[11,12]. However, the question itself is nuanced, depending not only on how one defines plate tectonics in general, but how, in detail, the interconnectivity, scale, depth, and thermal state of subduction zones may have changed through time[1,13].

Plate tectonics is characterized by relative motions between rigid lithospheric plates that are generated and destroyed along divergent (spreading ridges) and convergent (subduction zones) plate boundaries (Supplementary Fig. S1a), respectively. Recognizing lithological

and structural assemblages that are diagnostic of these end-member plate boundary settings, and characterizing in detail how these characteristics may have changed through Earth's history, are important avenues of research in our quest to understand the geodynamic evolution of our planet. The Neoarchean to Paleoproterozoic is one of the most transformative intervals in Earth history, witnessing globally-diachronous cratonization and emergence of large subaerial continental landmasses[14,15], and a huge increase in the amount of atmospheric oxygen (i.e., the Great Oxidation Event, GOE)[16]. This interval has been suggested to be the key period during which some form of plate tectonics was established globally, giving rise to the onset of the supercontinent cycle[1,10,17]. However, temporally- and spatially-associated divergent and convergent plate boundary rock assemblages and orogenic zonation (Supplementary Fig. S1b) that signify lateral plate motions have rarely been documented in the Archean rock record.

Here, we integrate petrology, geochemistry, and thermodynamic and trace element modeling of rocks from the Neoarchean Angou

[1]Badong National Observation and Research Station for Geohazards, State Key Laboratory of Geological Processes and Mineral Resources, Center for Global Tectonics, School of Earth Sciences, China University of Geosciences, Wuhan 430074, China. [2]School of Earth and Planetary Sciences, The Institute for Geoscience Research, Curtin University, Perth, WA 6102, Australia. [3]School of the Environment, University of Windsor, Windsor, ON N9B 3P4, Canada. ✉ e-mail: hbyjdg@cug.edu.cn; tkusky@gmail.com

Complex in the southern North China Craton. We identify temporally- and spatially-associated lithostructural belts that contain mid-ocean ridge–passive margin and intra-oceanic supra-subduction-zone arc/ forearc complexes. We contend that these associations formed at divergent and convergent margin settings, respectively, then were juxtaposed during accretionary-to-collisional orogenesis, thereby providing direct evidence for plate tectonic processes involving sea-floor spreading, subduction initiation, subduction accretion, arc magmatism, and arc–continent collision at the end of the Archean. We further use quantitative thermodynamic forward modeling to char-acterize the thermal state of the subduction zone. Finally, we explore the plate tectonic style in the Neoarchean and link this with the shaping of Earth's late Neoarchean and early Paleoproterozoic surficial envir-onment (e.g., oceanic–atmospheric oxygenation), that paved the way to a planet habitable by multicellular life.

## Results and discussion
### Geological background of the Angou Complex
The North China Craton (NCC) comprises the Eastern and Western blocks and the intervening ~1600-km-long, N–S-trending Central Orogenic Belt, with several Paleoproterozoic tectonic belts in its interior and margins (Fig. 1a)[18,19]. The Central Orogenic Belt formed by accretion of an arc system or systems, followed by collision with the western margin of the Eastern Block during late Archean to earliest Paleoproterozoic orogenesis[18], with the widespread development of ca. 2.55–2.50 Ga arc magmatism[20], mélanges[21], and ca. 2.51–2.48 Ga metamorphism[22] (Supplementary Note 1, Supplementary Fig. S2a).

The Angou Complex is located in the southern segment of the Central Orogenic Belt of the NCC (Fig. 1b), and consists of Neoarchean tonalite–trondhjemite–granodiorite (TTG) gneisses, metamorphosed volcanic rocks of the basalt–andesite–dacite–rhyolite suite and sub-marine sedimentary sequences, with minor mafic, dioritic, and granitic plutons or dikes (Supplementary Figs. S3–S9)[23,24]. Based on differences in lithologic association, structural style, and whole-rock geochemical composition, the Angou Complex can be subdivided into three lithostructural belts that are juxtaposed along faults/shear zones (Fig. 1b, c): (1) the Eastern Belt, comprising metabasalt and fine-grained metasedimentary sequences (Supplementary Figs. S3–4); (2) the Central Belt, composed of metamorphosed mafic to felsic volcanic rocks and sedimentary successions (Supplementary Figs. S5–6); and (3) the Western Belt, consisting of TTG gneisses and metabasites (Supplementary Fig. S7). All rocks were intruded by ca. 2.51–2.50 Ga and younger felsic and mafic plutons and/or dikes, and are uncon-formably overlain by the early Paleoproterozoic (<2.45 Ga) Songshan Group (Fig. 1b). These relationships indicate that tectonic juxtaposi-tion of the different lithostructural belts occurred at or before ca. 2.45 Ga. Based on our new fieldwork and laboratory analysis, we describe in detail the various components of the Angou Complex, including their whole-rock geochemistry and, where applicable, geo-chronology (see Methods, Supplementary Data 1–3 and Notes 2–3, and Supplementary Figs. S9–11).

### The Eastern Belt of the Angou Complex
The Eastern Belt of the Angou Complex is composed of a suite of metabasalt–chert–shale–banded iron formation (BIF) and well-bedded metasiltstone–shale sequences, with minor limestone (Supplementary Figs. S3 and S4). The metabasalts preserve massive, pillow, or vesicular structures (Supplementary Fig. S3a–d). The pillows are elongated and range from several centimeters to more than one meter in length (Supplementary Fig. S4). In places, thin bands of chert and/or carbo-nate occur along the chilled margins of the pillows. Locally, the pillow basalts are characterized by intense epidotization, reflecting seafloor hydrothermal alteration. Rare fine-grained mafic dikes intrude the metabasalt (Supplementary Fig. S4g). With the exception of BIF, the observed sequence of basalt–chert–shale–BIF ± limestone is

analogous to a typical Phanerozoic ocean plate lithostratigraphic assemblage recording migration of oceanic crust from the spreading ridge to the trench[25]. Within the Eastern Belt, this ocean plate strati-graphy is in thrust contact with a ~2-km-thick sequence of metamor-phosed well-bedded quartz–mica schist, quartz schist, and minor BIF (Supplementary Fig. S3e–h), which is interpreted as a fine-grained submarine siliciclastic (mainly silty and pelitic) and chemical sedi-mentary succession that formed along a relatively stable continental margin. The foliation within the metabasalt and metasedimentary sequences mostly dips at moderate angles towards the W or SW (Fig. 1b).

Zircons from a thin felsic volcanic layer (21AG02-2) and a dacite porphyry (22AG04) yield weighted mean U–Pb ages of $2523 \pm 13$ Ma (2σ) and $2527 \pm 17$ Ma, respectively (Supplementary Fig. S11a, b), with positive $\varepsilon_{Hf}(t)$ values between +3.4 and +6.4. Zircons from a granite dike (22AG08) intruding the Eastern Belt yield a weighted mean $^{207}Pb/^{206}Pb$ age of $2504 \pm 15$ Ma (Supplementary Fig. S11c), and posi-tive $\varepsilon_{Hf}(t)$ values (+3.2 to +4.2). In terms of whole-rock geochemistry, basaltic rocks in the Eastern Belt are tholeiitic (Fig. 2a, b) with vari-able concentrations of $SiO_2$ (47.35–52.91 wt%), and $TiO_2$ (1.05–1.92 wt %), with moderate Mg# [$100 \times$ atomic Mg/(Mg + Fe$^{2+}$)] values (44–58, except for five low-MgO samples) (Supplementary Data 1). They are characterized by depleted to nearly-flat chondrite-normalized rare earth element (REE) patterns (La/Sm$_{cn}$ = 0.67–0.96, cn refers to normalization to chondrite[26]), lack prominent high-field strength element (HFSE) anomalies (Fig. 2c, d), and are compositionally similar to average Phanerozoic mid-ocean ridge basalt (MORB)[26], consistent with their depleted Nd isotopic compositions ($\varepsilon_{Nd}(t) =$ +2.98 to +4.24)[27]. On a Sm/Yb versus La/Sm diagram (Fig. 3a), the metabasalts have compositions consistent with low-degree melting of spinel peridotite. Trace element diagrams (Fig. 3b, c) support a mid-ocean ridge (MOR) origin. Based on the field and geochemical data, we suggest that the Eastern Belt of the Angou Complex initially formed at a spreading ridge and an adjacent con-tinental margin.

### The Central and Western belts of the Angou Complex
The Central Belt of the Angou Complex consists predominantly of metamorphosed volcano-sedimentary assemblages (Supplementary Fig. S5), with a dominance of basaltic rocks (now amphibolites) in the west (~4 km thick), and of intermediate to felsic volcanic rocks and sedimentary rocks in the east (~2 km thick, Fig. 1b). The volcanic rocks comprise (meta-) basalt, high-Mg basalt, and andesite–dacite–rhyolite (ADR), and the sedimentary successions consist of (meta-) chert, shale and minor BIF. The strongly-foliated lower-amphibolite facies meta-basites dip at moderate to steep angles towards the SW (Fig. 1b). The intermediate and felsic volcanic rocks show evidence of top-to-the-NE shearing, with asymmetric feldspar porphyroclasts in dacites (Sup-plementary Fig. S8g–i), and deformed volcanic breccia (Supplemen-tary Fig. S5e). The chert, shale, and BIF preserve parallel bedding, consistent with a deep-marine origin (Supplementary Fig. S5g, h).

Zircon grains from three samples of felsic volcanic rock (18AG05-3, 18AG03-1 and 18AG06-4) from the Central Belt exhibit well-developed oscillatory zoning in cathodoluminescence (CL) images (Supplementary Fig. S10) and yield upper intercept zircon U–Pb ages of $2537 \pm 19$ Ma, $2529 \pm 24$ Ma, and $2525 \pm 23$ Ma (Supplementary Fig. S11d–f), respectively. Zircons from the felsic volcanic rocks have positive $\varepsilon_{Hf}(t)$ values between +4.0 and +7.4 (average +5.8). Zircon grains from a metapelite (18RZ03-1) interbedded with metabasalt yield a weighted mean $^{207}Pb/^{206}Pb$ age of $2533 \pm 25$ Ma (Supplementary Fig. S11g), interpreted as the maximum depositional age of its silici-clastic protolith. Zircons from a K-rich granite dike (18AG08-4) intruding the metavolcano-sedimentary assemblages of the Central Belt yield an upper intercept U–Pb age of $2499 \pm 36$ Ma (Supplemen-tary Fig. S11h).

The Western Belt consists mainly of TTG gneisses and amphibolites (Fig. 1b), with minor mafic and felsic dikes (Supplementary Fig. S7). The TTG gneiss is highly deformed and migmatitic, cross-cut by abundant leucogranitic dikes (Supplementary Fig. S7b–d). The gneiss is composed dominantly of plagioclase, quartz and minor biotite, with accessory titanite, apatite and zircon (Supplementary Fig. S8a). Most are trondhjemitic, with high concentrations of $SiO_2$ (71.72–77.19 wt%) and $Na_2O$ (5.03–6.4 wt%), and high Sr/Y (163–256) and $La/Yb_{cn}$ (26–47) ratios, and highly-fractionated REE patterns and positive Eu/Eu* anomalies [1.38–2.15, Eu/Eu* = Eu/√(Sm*Gd), Fig. 2g].

Mafic lenses and boudins range from several meters to tens of meters in width (Fig. 1b and Supplementary Fig. S7a) and consist mainly of amphibole with minor plagioclase (Supplementary Fig. S8b). Leucogranite dikes are weakly-deformed and contain mostly plagioclase and quartz. Zircons from an amphibolite sample (22RZ06b) that is associated with trondhjemitic gneisses (Supplementary Fig. S7a) have core–rim textures (Supplementary Fig. S10) consistent with magmatic and metamorphic origins, respectively. The magmatic (mostly cores) and metamorphic zircons yield weighed mean $^{207}Pb/^{206}Pb$ ages of 2549 ± 18 Ma and 2483 ± 28 Ma (Supplementary Fig. S11i), interpreted

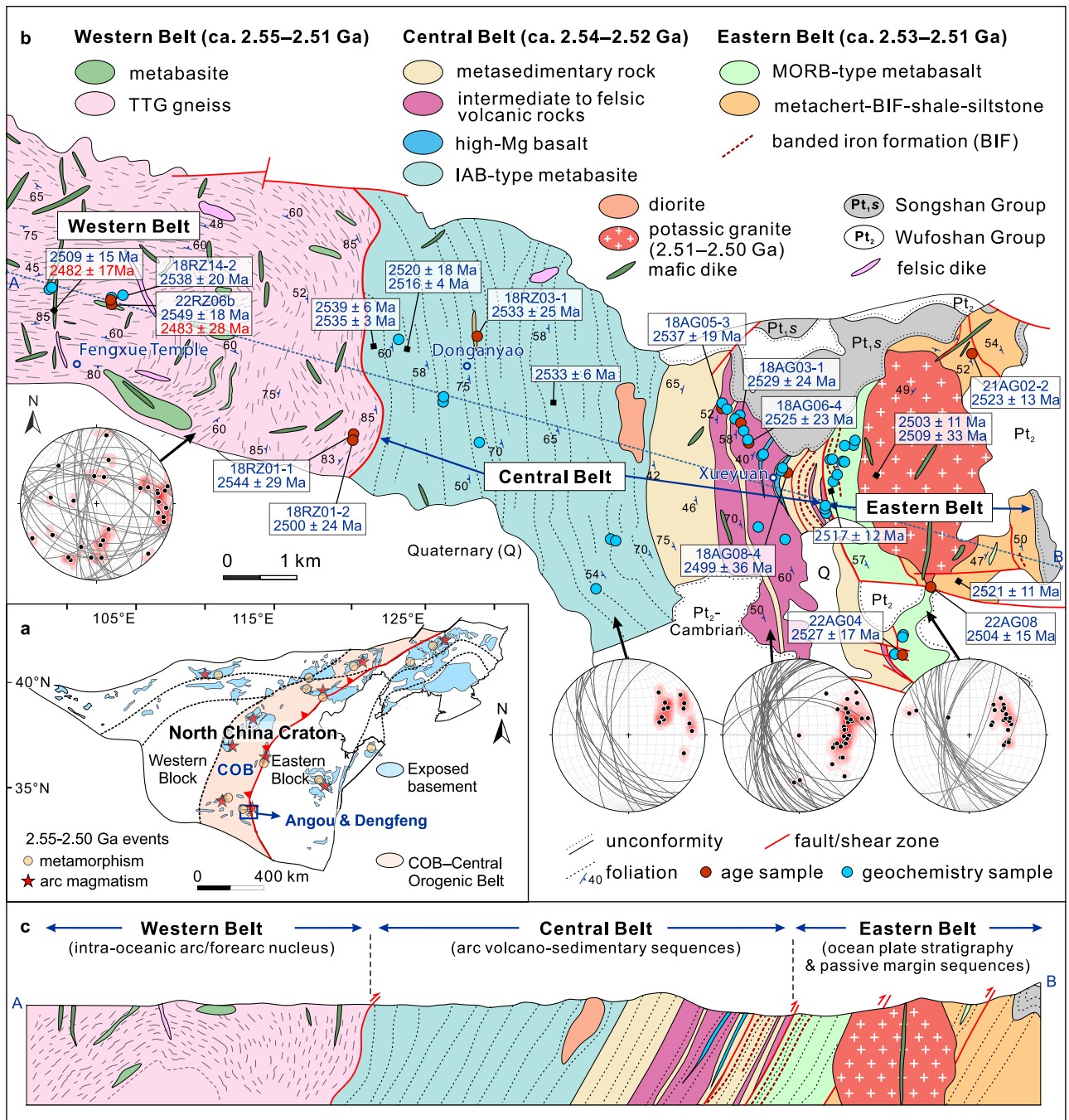

**Fig. 1 | Geological maps and lithological-structural relations. a** Simplified tectonic map showing the tectonic framework of the North China Craton (NCC, modified from refs. 18,70). **b** Geological map of the Angou Complex in the southern NCC (modified from ref. 23). Three lithostructural belts are shown: the Western, Central and Eastern belts, along with new and previously-reported age data (see Supplementary Data 8). The foliations of the different units are plotted in lower hemisphere equal area stereographic projections. **c** Lithostructural cross-section of the Angou Complex. TTG: tonalite–trondhjemite–granodiorite; MORB: mid-ocean ridge basalt; IAB: island arc basalt.

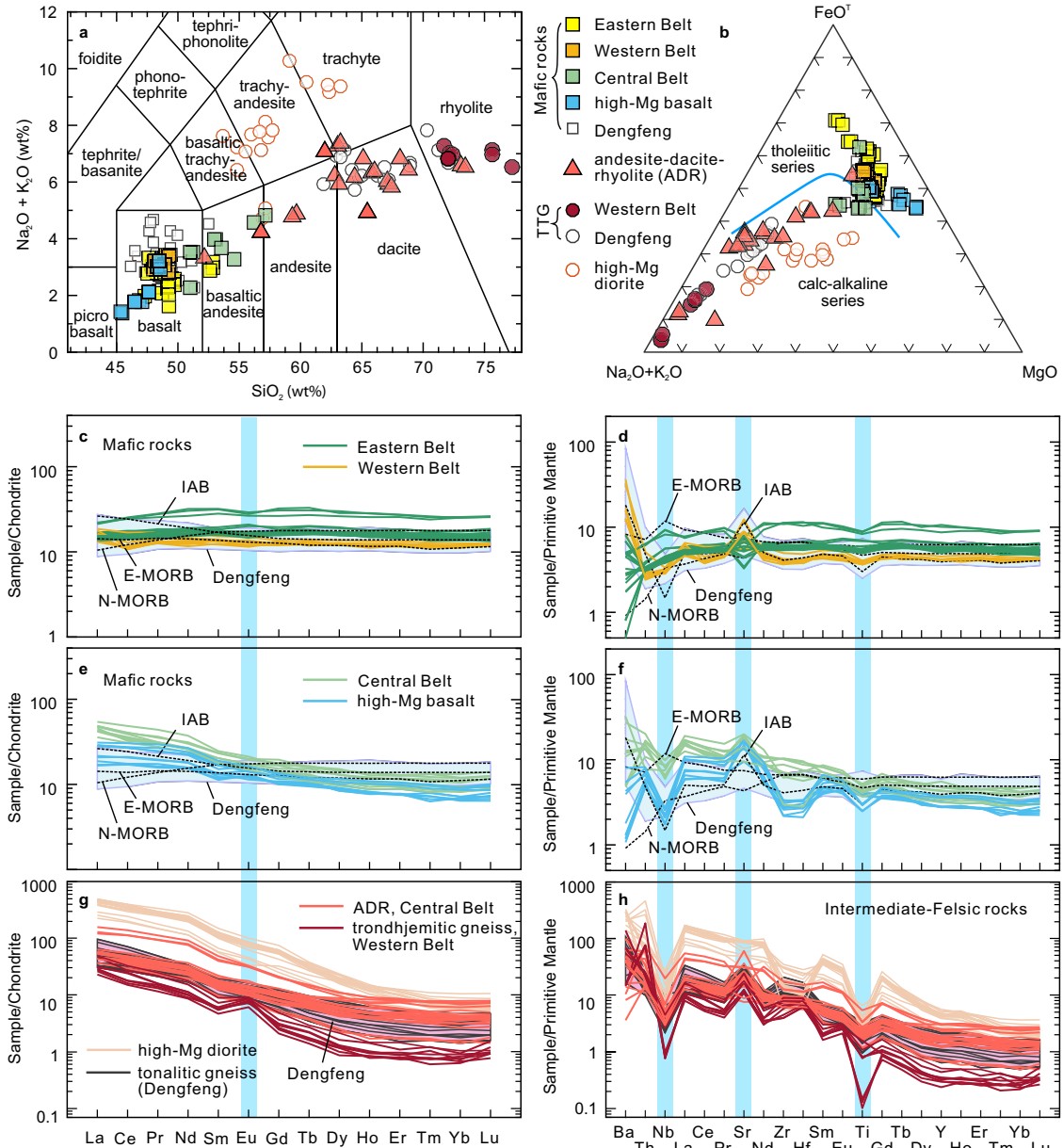

**Fig. 2 | Major and trace element geochemical characteristics of the Angou Complex. a** Na$_2$O + K$_2$O versus SiO$_2$ diagram. **b** AFM (Na$_2$O + K$_2$O–MgO–FeO$^T$) diagram. **c**, **e**, **g** Chondrite normalized rare earth element diagrams. **d**, **f**, **h** Primitive mantle-normalized trace element diagrams. Results from the adjacent and coeval Dengfeng Complex that preserves pristine tonalitic TTG gneisses, high-Mg diorite

(sanukite), and arc/forearc metabasalts are compiled and plotted for comparison and discussion. Normalized values of chondrite and primitive mantle, N-MORB (normal mid-ocean ridge basalt), E-MORB (enriched mid-ocean ridge basalt), and IAB (island arc basalt) are from ref. 26.

to date crystallization of its magmatic protolith and high-grade metamorphism, respectively. Zircon grains from TTG gneiss samples (18RZ14-2 and 18RZ01-1) record upper intercept ages of 2538 ± 20 Ma and 2544 ± 29 Ma (Supplementary Fig. S11j, k), respectively, interpreted to date crystallization of their magmatic protoliths. Zircons from the gneisses have positive ε$_{Hf}$(t) values between +5.7 and +7.6 (average + 6.5). A leucogranitic dike (18RZ01-2) within the gneiss yields an upper intercept U–Pb zircon age of 2500 ± 24 Ma (Supplementary Fig. S11l), interpreted as the crystallization age.

Similar to basalts from the Eastern Belt, mafic rocks (ca. 2.55 Ga) from the Western Belt have MORB-like tholeiitic compositions (Fig. 2a, b), but are relatively enriched in some large-ion lithophile elements (LILEs, e.g., Ba and Sr) (Fig. 2d), and have higher Ba/Th ratios, indicating a minor component of slab-derived aqueous fluids in their mantle source[28] (Fig. 3c). Such characteristics are consistent with those

of forearc basalts formed during subduction initiation[28,29]. By contrast, mafic rocks (including high-Mg basalts and basaltic andesites) from the Central Belt have fractionated REE patterns (La/Sm$_{cn}$ = 1.31–2.68, Fig. 2e), are depleted in HFSEs (Nb, Ti, Fig. 2f) and enriched in LILEs, and preserve high Th/Nb ratios, features typical of island arc basalts (IAB) derived from a mantle source that interacted with a significant volume of slab-derived fluids and/or melts (Fig. 3b, c)[28].

The intermediate and felsic volcanic rocks in the Central Belt are dominated by dacite, with subordinate andesite and rhyolite (Fig. 2a). These rocks are enriched in LREE and LILE, depleted in heavy REE (HREE), exhibit negative HFSE anomalies (Fig. 2g, h), and have high Sr contents, Sr/Y (23–110, Fig. 3d) and La/Yb$_{cn}$ (5.6–39) ratios. Such characteristics are typical of adakites from circum-Pacific arcs[30,31]. On a Th/Yb versus Nb/Yb diagram (Fig. 3b), the MORB-like rocks from the Western Belt and high-Mg basalts from the Central Belt plot in the field

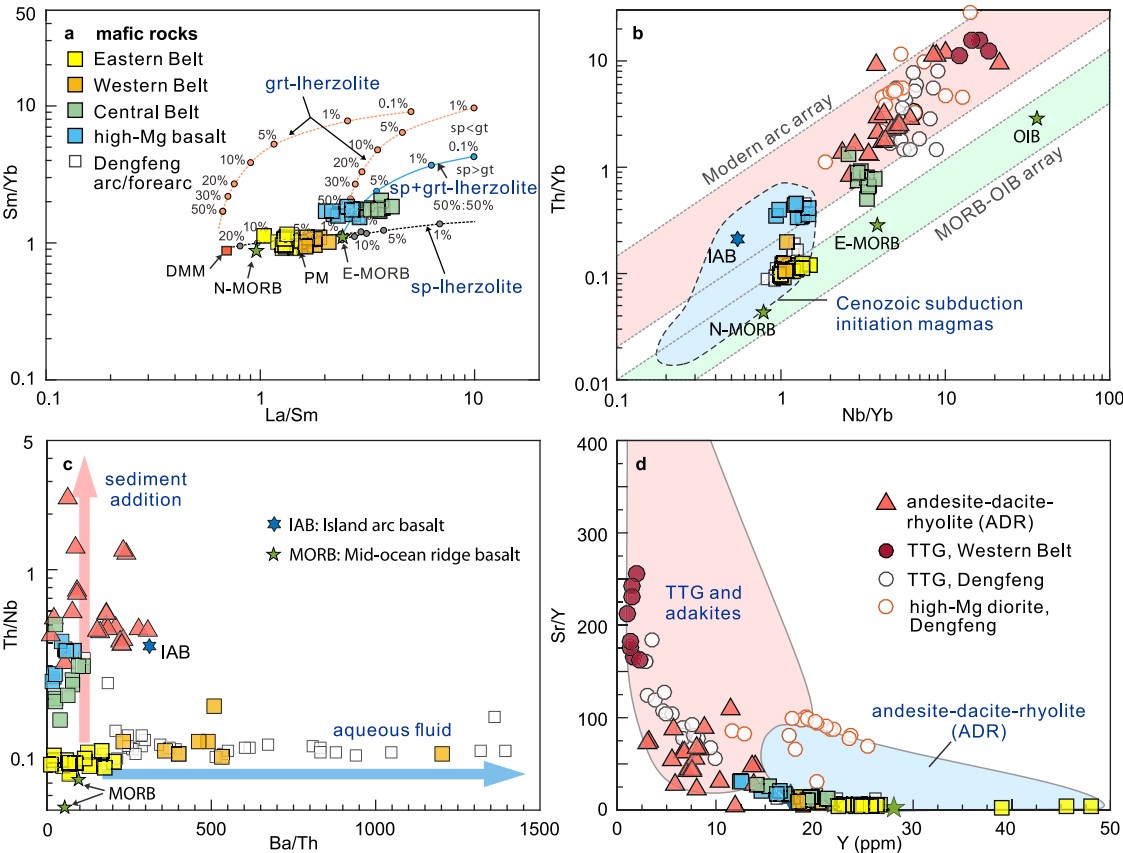

**Fig. 3 | Trace element plots distinguishing magma sources and tectonic settings. a** Sm/Yb versus La/Sm[71]. **b** Th/Yb versus Nb/Yb[72]. Data sources for Cenozoic subduction-initiation magmas are the same as ref. 29. **c** Th/Nb versus Ba/Th. **d** Sr/Y versus Y[30]. DMM depleted MORB mantle, PM primitive mantle, OIB oceanic island basalt.

of Cenozoic magmas formed during subduction initiation[29], whereas the IAB-like mafic rocks and adakitic volcanic rocks from the Central Belt plot within the arc array. We thus interpret the older mafic rocks (ca. 2.55 Ga) in the Western Belt and younger mafic to felsic volcanic rocks (ca. 2.54–2.52 Ga) in the Central Belt to represent intra-oceanic supra-subduction-zone (SSZ) forearc basalt and an IAB-ADR volcanic-arc suite, respectively.

**Thermodynamic and trace element modeling**

TTG and adakites are generally regarded as the product of partial melting of hydrous mafic rocks[32,33], reflecting either direct melting of subducting oceanic crust, island arcs, and/or oceanic plateaus, with or without interaction with the mantle wedge[30,34,35], or through anatexis near the base of thick or thickened arc, or plateau-like mafic crust[36,37]. Clearly, the bulk composition, including the availability of $H_2O$, and the thermal state of the crust vary between these settings[38,39]. To constrain these variables, we conducted thermodynamic and trace element modeling using plausible source compositions (see below and Methods).

Most felsic volcanic rocks in the Angou Complex are compositionally similar to ca. 2.53–2.51 Ga tonalitic rocks (Fig. 2a, b, g, h) in the adjacent and coeval Dengfeng Complex (Supplementary Note 1, Supplementary Fig. S2b), which are typical of an Archean TTG–dacite association[30]. However, relative to the Angou trondhjemitic TTG gneisses, the Dengfeng gneisses were less affected by crystal fractionation and (we argue) more reliably characterize the TTG source[40]. Consequently, we use a compiled geochemical dataset of volcanic rocks from the Angou Complex and tonalitic TTG gneiss samples from the Dengfeng Complex to constrain the geodynamic setting of TTG and adakitic volcanic rocks. For the felsic rocks within the Angou and

Dengfeng complexes, partial melting of thick plateau-like crust is unlikely as all mafic rocks are IAB- or MORB-like, with oceanic island and plateau basalts notably absent. In addition, no Neoarchean komatiite has been found in the southern NCC. In order to discriminate between the end-member models (partial melting of MORB versus IAB) for the genesis of the TTGs and adakitic rocks, we conducted thermodynamic and trace element modeling using average MORB (MORB-AV; $n = 21$) and IAB (IAB-AV; $n = 6$) compositions of the Angou Complex (Supplementary Data 4) as a function of the $P–T$ conditions appropriate to these two geodynamic settings (Figs. 4 and 5)[38,39].

For the mineral/melt partition coefficients used (ref. 36, Supplementary Data 5), our modeling shows that partial melting of LREE- and LILE-enriched average IAB at $P–T$ ranges of 750–950 °C and 1.2–2.5 GPa produces strongly REE-fractionated melts that are too enriched in LREE and Th, and too depleted in HREE to be the source rocks of the TTG and adakitic rocks (Supplementary Data 6 and Fig. 5b). This precludes models of partial melting of normal or thickened arc root or a subducting island arc. By contrast, low-degree (~7–15%) partial melting of an average MORB at 760–810 °C and 1.6–1.8 GPa, in which the high-pressure granulite to eclogite facies residual mineral assemblage consists of garnet–clinopyroxene–hornblende–rutile ±minor plagioclase (Supplementary Data 6 and Fig. 4a), produces melts with compositions that match well with those of the tonalitic and adakitic rocks in the Angou and Dengfeng complexes (Fig. 5a), supporting a petrogenesis through partial melting of subducting MORB-type oceanic crust.

Relative to those of the Dengfeng tonalitic gneisses[20], the Angou trondhjemitic gneisses have higher $SiO_2$ concentrations, lower total REE contents, more fractionated HREE patterns, more pronounced negative Nb–Ta anomalies, stronger positive Eu anomalies (Fig. 2g, h),

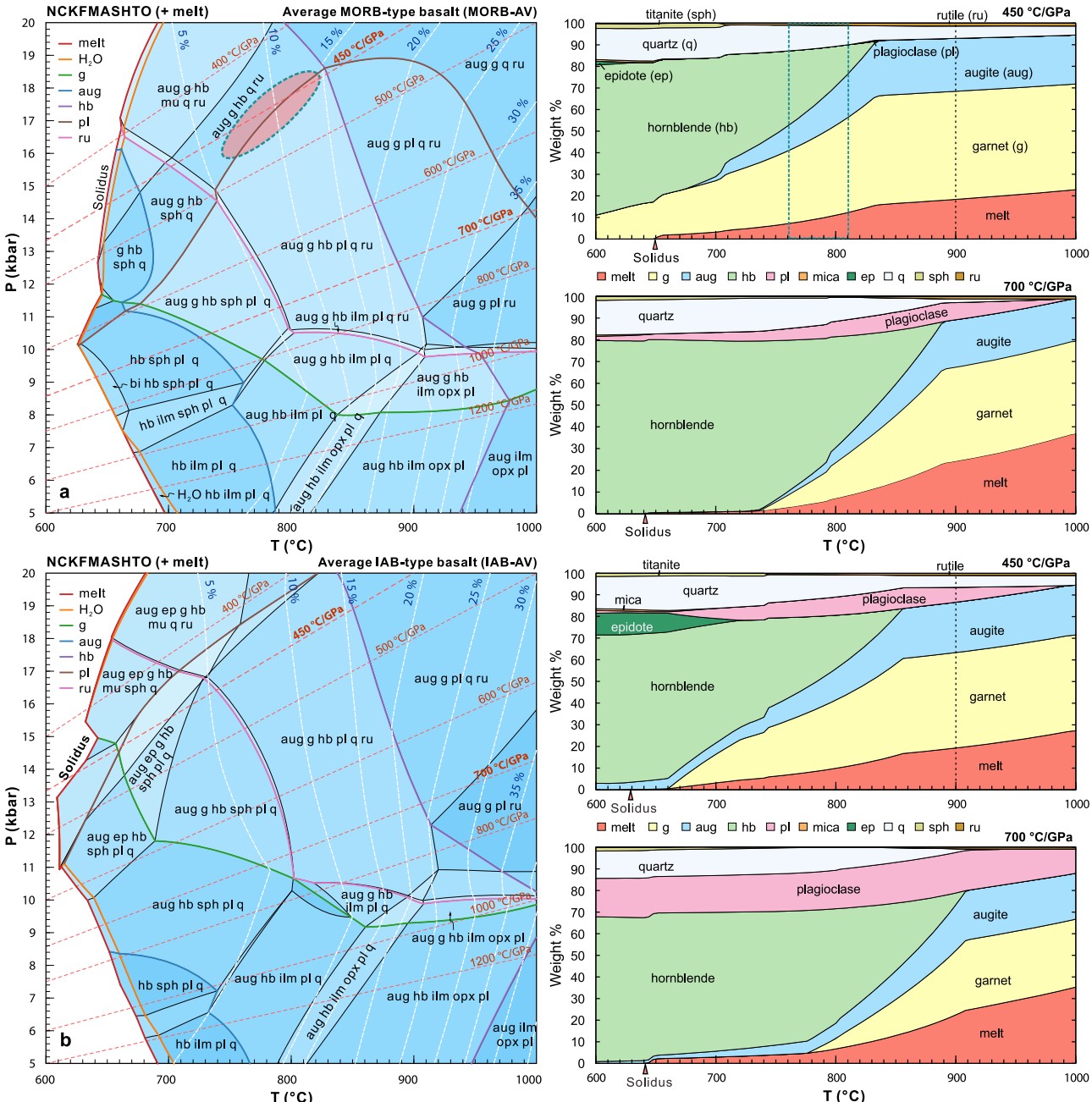

**Fig. 4 | Isochemical phase diagrams (pseudosections) and calculated proportions (wt.%) of phases along specified thermal gradients. a** Partial melting of an average MORB-type basalt (MORB-AV). **b** Partial melting of an average IAB-type basalt (IAB-AV). The white dashed lines and blue numbers in the phase diagrams indicate the proportions of melt. The red ellipse in panel **a** marks the P–T range that can generate melt matching the compositions of the Dengfeng tonalitic TTG. 1 GPa = 10 kbar.

and likely represent more highly-evolved compositions due to fractional crystallization of a tonalitic parental magma[40]. Trace element compositions modeled assuming non-modal Rayleigh fractional crystallization (Supplementary Data 7) and a minimum Dengfeng TTG composition show that the Angou trondhjemitic rocks can be explained by 20–30% fractional crystallization dominated by hornblende (70–80%), with subordinate plagioclase, biotite, and accessory ilmenite (± zircon, apatite) (Fig. 5a).

In summary, our modeling results support a model of partial melting of subducted MORB-like oceanic crust, accompanied by variable fractional crystallization of the melts/magmas so produced, in the genesis of the adakitic and TTG rocks in the Angou and Dengfeng complexes.

## Neoarchean seafloor spreading and plate convergence
In the Central and Western belts of the Angou Complex, basaltic and felsic magmatic rocks have structural, petrological, and geochemical characteristics consistent with formation in a SSZ arc/forearc setting. In Phanerozoic intra-oceanic subduction systems, from the base up, the arc–forearc crust consists mainly of MORB-like igneous rocks (e.g., gabbro, forearc basalt), boninite/high-Mg andesite/intermediate and felsic plutons and/or dikes (e.g., diorite, TTG), and younger volcanic (e.g., IAB, ADR) and submarine sedimentary/volcaniclastic rocks[41,42]. Though not always observed and/or preserved in ancient orogenic belts, the forearc basalt–boninite suite is commonly regarded as an indicator of subduction initiation associated with forearc spreading, whereas the evolved TTG, IAB, and ADR suites represent subsequent

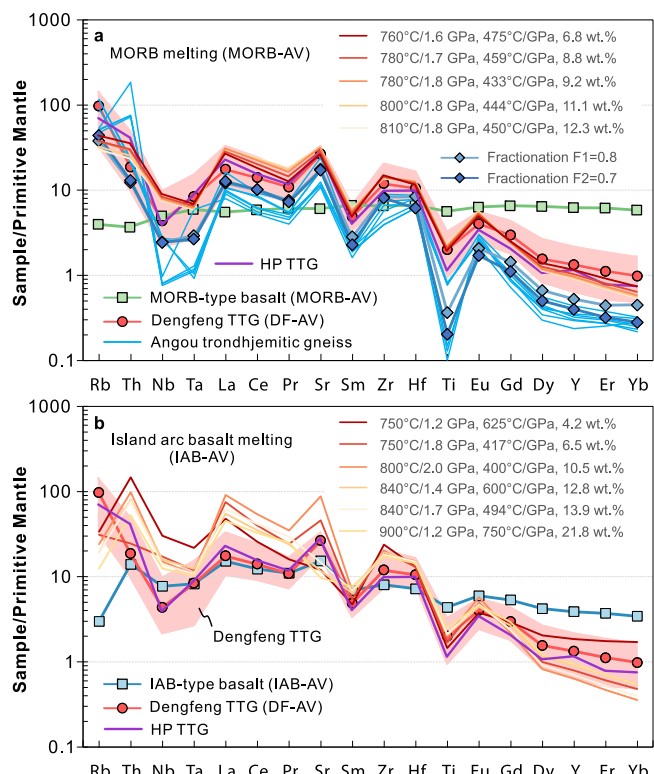

**Fig. 5 | Trace element modeling. a** Using average MORB-type basalt (MORB-AV) as the source rock. **b** Using average IAB-type basalt (IAB-AV) as the source rock. Full modeling results are presented in Supplementary Data 6, and selected representative results are shown here. The compositions of evolved melts resulting from removal of two assumed fractionating mineral assemblages (Supplementary Data 7) are shown. Primitive mantle normalized elemental compositions are from ref. [26]. The average composition of high-pressure (HP) TTG is from ref. [32].

mature arc magmatism[28,42,43]. In the Western and Central belts of the Angou Complex, although boninites are absent, the ca. 2.55–2.51 Ga forearc basalt–IAB–high-Mg basalt/diorite–ADR suite, and adakitic TTG gneisses are broadly akin to Phanerozoic intra-oceanic arc–forearc complexes (e.g., circum-Pacific Izu-Bonin–Mariana, and Tonga). We conclude that such complexes record a magmatic progression from subduction initiation (ca. 2.55 Ga) to subsequent maturation (ca. 2.54–2.51 Ga) of a Neoarchean intra-oceanic subduction system along a convergent paleo-plate boundary.

The Eastern belt of the Angou Complex contains MORB-like basalts, with minor chert–shale–BIF layers, and lenses of limestone. Although the MORB-like basalts could have formed at a mid-oceanic ridge or at spreading centers in the backarc or forearc, based on the following lines of argument, we suggest they most likely formed at a mid-ocean ridge, then were emplaced during subduction-accretion. (1) The MORB-like unit formed at ca. 2.53–2.51 Ga, significantly later than the ca. 2.55 Ga forearc basalts but coeval with the ca. 2.54–2.51 Ga volcano–plutonic rocks of island arc-affinity from the Western–Central belts of the Angou Complex and the arc/forearc unit of the adjacent Dengfeng Complex (Supplementary Fig. S13); this violates the 'subduction initiation rule'[42], in which forearc basalt should be the earliest phase of subduction-related magmatism. (2) The ca. 2.55–2.51 Ga mafic to felsic volcano-plutonic rocks in the Central and Western belts constitute a typical sequence of island arc/forearc complexes, whereas the MORB-like unit in the Eastern Belt is characterized by repeated ocean plate stratigraphic sequences, and was more likely accreted to the western arc/forearc margin during subduction–accretion rather than obducted. Collectively, the MORB-like unit is consistent with typical lithological associations of MOR-type ocean plate stratigraphy that

record the history of ocean crust as it travels from a spreading ridge to a trench. Seafloor spreading generated MORB-type basalts and minor diabase dikes, with subsequent seawater alteration causing epidotization in the pillow basalts (Supplementary Fig. S4).

The ocean plate stratigraphy was thrust over a ~2-km-thick submarine quartz-mica schist–quartz schist–BIF sequence (Supplementary Fig. S3e–h), which is interpreted as a stable continental margin assemblage deposited in relatively deep water, probably on a continental shelf-to-slope environment in the Eastern Block of the Proto-North China Craton. To the east of this sequence, the basement rocks are dominated by >2.66–2.55 Ga TTG gneisses and minor amphibolites, which have been regarded as the basement in the western margin of the Eastern Block[44]. The sedimentary succession (ca. 2.54–2.51 Ga) deposited on the older continental margin of the Eastern Block is thus consistent with sedimentation along a passive continental margin. This discontinuous Neoarchean passive margin sedimentary sequence comprising submarine siliciclastic and/or carbonate rocks can be traced for at least 400 km from Angou to Zanhuang along the western margin of the Eastern Block[18,19] (Fig. 6a), and is one of the oldest preserved and laterally extended passive margin sequences on Earth[19,45]. Thus, field, geochronological and geochemical data collectively indicate that the Eastern Belt of the Angou Complex contains lithological associations closely corresponding to a MOR-type ocean plate stratigraphy and a passive continental margin sequence, which together record extensional tectonics leading to seafloor spreading at a mid-oceanic ridge and subsequent sedimentation during thermal subsidence along a continental margin.

The temporally- and spatially-associated plate boundary rock assemblages in the southern NCC are best explained by plate tectonic processes that involved seafloor spreading, intra-oceanic subduction initiation, arc maturation (2.55–2.51 Ga) and final arc–continent collision (2.51–2.45 Ga) of the Proto-North China Craton (Fig. 6). At around 2.55–2.52 Ga, a divergent paleo-plate boundary was characterized by seafloor spreading outboard of a passive margin along the western margin of the Eastern Block (Fig. 6a). A new subduction zone was initiated at ca. 2.55 Ga, leading to the development of an intra-oceanic arc/forearc system (ca. 2.55–2.51 Ga Central arc system) (Fig. 6a). Forearc basalts formed first by decompression melting of a mantle that had undergone limited metasomatism by slab-derived fluids during the subduction initiation stage[28,42]. Continuous intra-oceanic subduction and slab roll-back led to arc maturation, with the generation of typical SSZ magmatic sequences, including an intrusive TTG suite and a high-Mg basalt/diorite (sanukitoid)–IAB–ADR volcanic suite in the western Angou and Dengfeng regions, as well as in other segments of the Central arc system[20,44,46] (Fig. 6b). These rocks were produced through partial melting of young and warm subducting oceanic crust and the overlying mantle wedge, with an enhanced contribution from slab-derived fluids and melts[42,47]. The subduction processes documented here resemble those in the Izu–Bonin–Mariana intra-oceanic subduction system, albeit with warmer subduction zone geotherms as revealed by our modeling, which shows that the TTG and adakitic magmas were generated under subduction zone geotherms of ~440–470 °C/GPa (Fig. 7).

The duration of arc magmatism indicates that subduction continued for about 40 million years (Myrs) from ca. 2.55 to at least 2.51 Ga (Supplementary Data 8, Supplementary Fig. S13). Assuming a modest absolute plate velocity of 25 km/Myr and a duration of subduction of ~40 ± 10 Myrs, this indicates subduction of >1000 ± 250 km of oceanic lithosphere (Fig. 6b). Continuous plate convergence led to accretion of the Central arc terrane(s) onto the western margin of the Eastern Block of the NCC to form the ~1600 km Central Orogenic Belt during the late Neoarchean to early Paleoproterozoic (ca. 2.51–2.45 Ga) (Fig. 6c), resulting in the Neoarchean ocean plate stratigraphy/ophiolitic mélanges, arc magmatic rocks, and paired metamorphism (Figs. 1a and 6a)[18,20,22,46,48]. These features are similar to the structural,

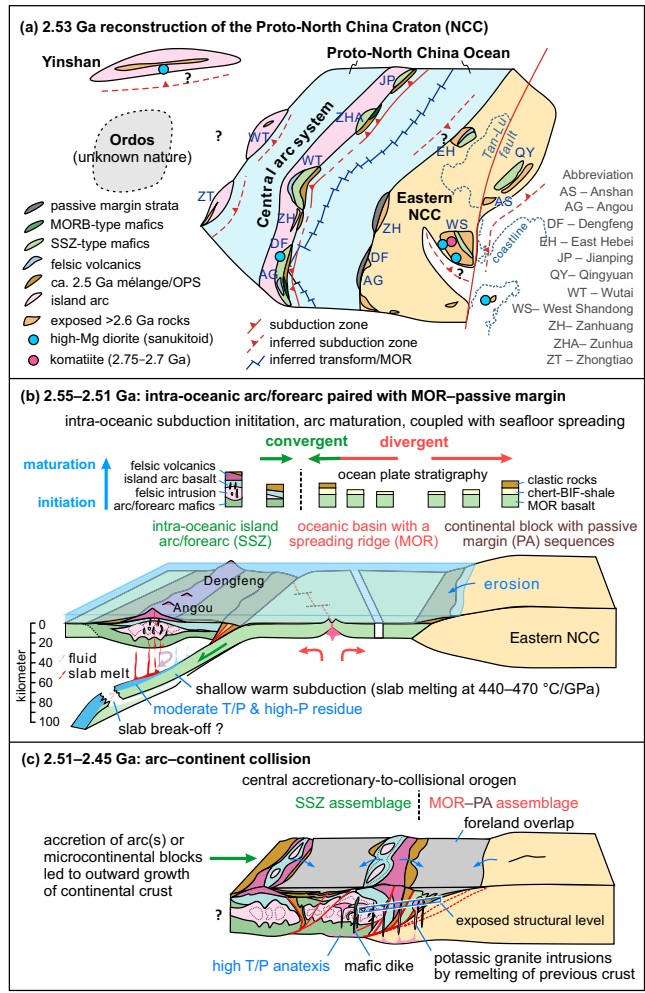

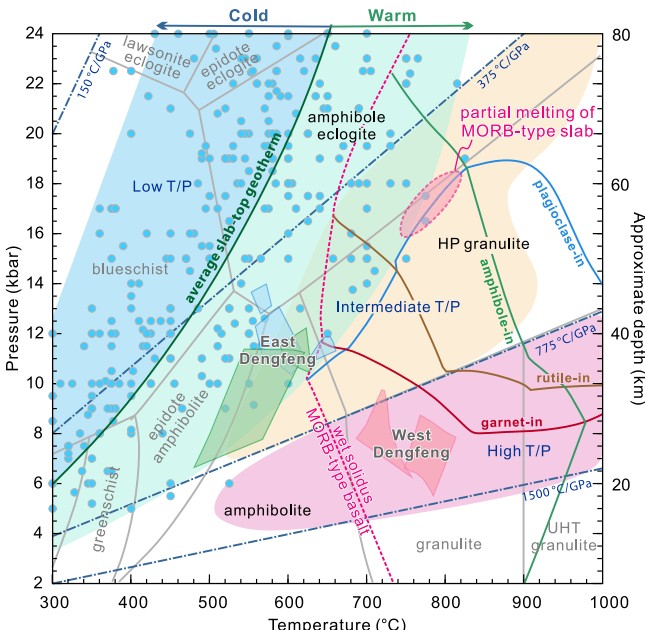

**Fig. 7 | Partial melting P–T conditions in the Angou Complex.** Phase diagram showing that the melting P–T conditions for generating adakitic magmas partially overlap with the slab-top P–T conditions (blue and green areas) from Neoproterozoic to present subduction zones, reconstructed from exhumed metamorphic rocks (blue circles) in orogenic belts[51] (modified from ref. [22]). The boundaries of wet solidus and key phases (garnet, amphibole, plagioclase, and rutile) are outlined, and were calculated based on the average MORB protolith (MORB-AV) in the Angou Complex. Note that cold and average slab-top geotherms do not intersect the wet solidus of basalt. The P–T conditions of the Archean Dengfeng paired metamorphic belt are also shown[22].

**Fig. 6 | Reconstructed late Neoarchean tectonic model for the southern North China Craton. a** Palinspastic reconstruction of the Proto-North China Craton at ca. 2.53 Ga. Subduction-related geological records are widely distributed along the Central arc system[18,19,44]. Passive continental margin sequences were deposited discontinuously along the southwestern margin of the Eastern North China Craton (Eastern NCC, or the Eastern Block)[18,19,44]. The geological units have been restored across the sinistral Tanlu fault following ref. [73]. **b** 2.55–2.51 Ga: Intra-oceanic subduction initiation and arc maturation in the west, paired with a subducting slab that originated from a spreading mid-ocean ridge, and with passive margin sedimentation in the Angou–Dengfeng segment of the Proto-North China Craton. **c** 2.51–2.45 Ga: The intra-oceanic arc/forearc(s) were accreted onto the western margin of the southeastern NCC, leading to the formation of the structurally juxtaposed mid-ocean ridge (MOR)–passive margin (PA) and supra-subduction zone (SSZ) belts in the Angou Complex. This marks an episode of orogenesis in the Central Orogenic Belt and a spreading–subduction–accretion-collisional Wilson Cycle-like plate tectonic process.

magmatic, and metamorphic styles observed in Phanerozoic accretionary orogens[49]. Widespread 2.51–2.45 Ga metamorphism, emplacement of 2.51–2.40 Ga K-rich granitoids, and the unconformable deposition of Paleoproterozoic foreland sedimentary sequences along the newly-assembled Central Orogenic Belt, define the history of Neoarchean accretion-to-collision orogenesis in the NCC, contemporaneous with the formation of ~2.6–2.5 Ga supercratons or the Kenorland supercontinent[14,50].

## Operation of plate tectonics in the Neoarchean
The defining feature of plate tectonics is relative lateral motions between rigid lithospheric plates concentrated within an interconnected network of transform (transform faults), divergent

(spreading ridges) and convergent (subduction zones) plate boundaries[2] (Supplementary Fig. S1a). In geological history, these processes are recorded by diagnostic rock assemblages in ancient orogens, such as passive margin sequences, MOR-type oceanic fragments, SSZ-type magmatic rocks, accretionary complexes, metamorphic rocks with bimodal thermobaric ratios (T/P), and forearc/foreland basin sequences, which are generally preserved and temporally and spatially associated, forming tectonic zonation across an orogen[18,49] (Supplementary Fig. S1b).

We interpret the studied rocks in the southern NCC to represent a rare example of a temporally- and spatially-associated Neoarchean MOR–passive margin and SSZ rock associations that were tectonically juxtaposed during arc–continent accretion-to-collision orogenesis (Fig. 6). Such lower- and upper-plate lithostructural associations occurring as discrete tectonic zones in a single orogenic belt allow for palinspastic reconstruction of the original ocean basin (which we term the Proto-North China Ocean, Fig. 6a). The data are consistent with a system of mid-ocean ridges and subduction zones/trenches in the late Neoarchean (ca. 2.55–2.51 Ga), during which subduction ultimately led to the closure of a >1000 (±250)-km-wide and ~1600-km-long oceanic basin (Fig. 6a), providing direct evidence for large-scale plate motions involving seafloor spreading and plate convergence, and thus the operation of mobile-lid plate tectonics in the late Neoarchean.

Our results suggest that the processes of late Neoarchean seafloor spreading, subduction initiation, arc maturation, and arc–continent collision were broadly similar to modern plate tectonic systems, albeit with elevated geotherms that permitted partial melting of the downgoing slab. The estimated thermal gradient (~440–470 °C/GPa) of slab melting is higher than Phanerozoic average slab-top geotherms, but overlaps with those of warm (generally young) subduction zones[51] (Fig. 7). Considering an average crustal density and no significant

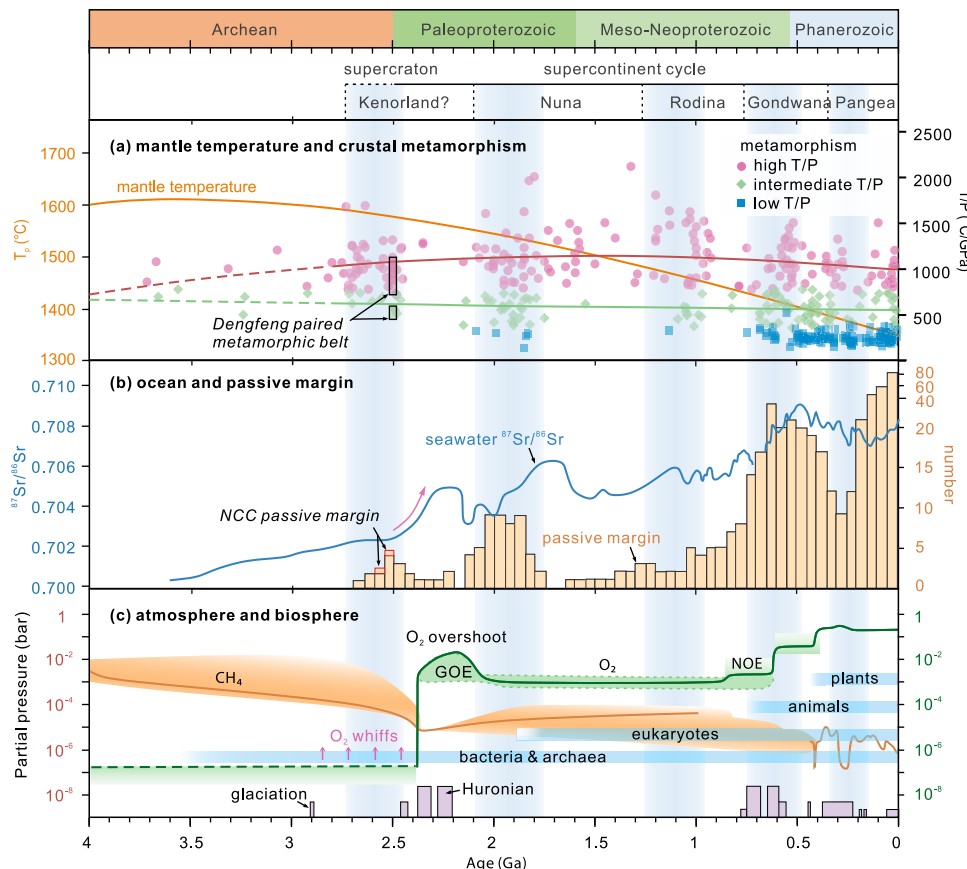

**Fig. 8 | Geological proxies for the secular evolution of mantle temperature, crustal metamorphism, ocean, atmosphere and biosphere. a** Mantle temperature and crustal metamorphism[1,55]. The ca. 2.5 Ga Dengfeng paired metamorphic belt[22] in the Central Orogenic Belt of the southern NCC is identified. **b** Seawater [87]Sr/[86]Sr isotopes[61], and the age distribution of passive margins[45]. Note that the Neoarchean passive margin in the NCC represents one of the oldest passive margins on Earth[45]. The red arrow highlights a significant increase of seawater [87]Sr/[86]Sr ratios during the early Paleoproterozoic. **c** Atmospheric oxygen ($O_2$) and methane ($CH_4$) levels relative to the present atmosphere[74], the evolution of life[75], and the distribution of glaciations[76]. The supercontinental cycles are outlined, with shaded light blue columns representing the collision phase of supercraton/supercontinent assembly[17]. GOE Great Oxygenation Event, NOE, Neoproterozoic Oxygenation Event.

tectonic overpressure, our modeled pressures of slab melting correspond to melting depths of up to 60 km, comparable with Phanerozoic examples of shallow flat-subduction of young oceanic crust (typically 60–80 km depth)[52]. The metamorphic *P–T* conditions recovered from the ca. 2.52–2.5 Ga Dengfeng paired metamorphic belt[22] are similarly consistent with warm subduction zones in the late Neoarchean.

Warm subduction zone thermal gradients reconstructed here can be attributed to the subduction of young oceanic slabs (generally with a comparatively short lifespan of ~30–50 Myrs) and higher mantle temperature in the Archean, which would have facilitated the melting of the mantle wedge and downgoing slab to generate mafic to felsic arc–forearc crust (Fig. 6b)[30]. MORB-type slab melting produces amphibolite to eclogite facies residues (Fig. 7), some of which were locally offscraped and exhumed to form relatively-coherent ocean plate stratigraphic sequences and block-in-matrix ophiolitic mélanges in warm subduction channels[22]. Partial melting and melt loss would have enhanced the negative buoyancy of the downgoing slab, providing impetus to the subduction system.

We further conducted independent estimates of mantle potential temperature ($T_p$) using the major element compositions of the basaltic rocks (following ref. 53, Supplementary Note 4, Data 9, and Supplementary Fig. S14). Our results show that the arc/forearc mafic rocks in the Western and Central belts have $T_p$ similar to those in the Izu-Bonin-Mariana arc/forearc[47,54], whereas MORBs in the Eastern Belt of the Angou Complex record $T_p$ of ~1410–1500 °C (average ~1450 °C), which is up to ~100 °C higher than for modern average

MORB mantle (~1350 °C)[53]. This broadly coincides with the interpreted emergence of bimodality in the *T/P* of crustal metamorphism[55] (Fig. 8a), the scarcity of komatiites <2.6 Ga[56], and recent thermal-mechanical modeling[57], all of which are consistent with secular cooling of the mantle. Since the Paleoproterozoic, the proportion of cold subduction zones has gradually increased, as has the strength of down-going slabs, perhaps permitting deeper and more continuous subduction. In most cases, cold subducting slabs dehydrate before they can melt (Fig. 7), hampering the generation of TTG and adakitic magmas in post-Archean times. By the middle Paleoproterozoic (ca. 2.2–1.9 Ga), widespread high-pressure–low-temperature metamorphic rocks, large-scale collisional orogenesis, and amalgamation of the first widely-accepted supercontinent signify the latest timing for the establishment of the 'modern-style' plate tectonic regime, as many defined, characterized by the occurrence of cold and deep subduction/collision and globally-interconnected network of narrow plate boundaries[1,10,58].

## Implications for Neoarchean–Proterozoic surficial environments

The plate tectonic processes, including seafloor spreading, subduction, and arc/micro-continent collision may have become globally widespread at least by the late Neoarchean–early Proterozoic, leading to the growth of continental landmasses and ultimately cratons, supercratons, and even a supercontinent[14,15,50]. This geodynamic reorganization likely caused profound changes in the deep Earth, but also

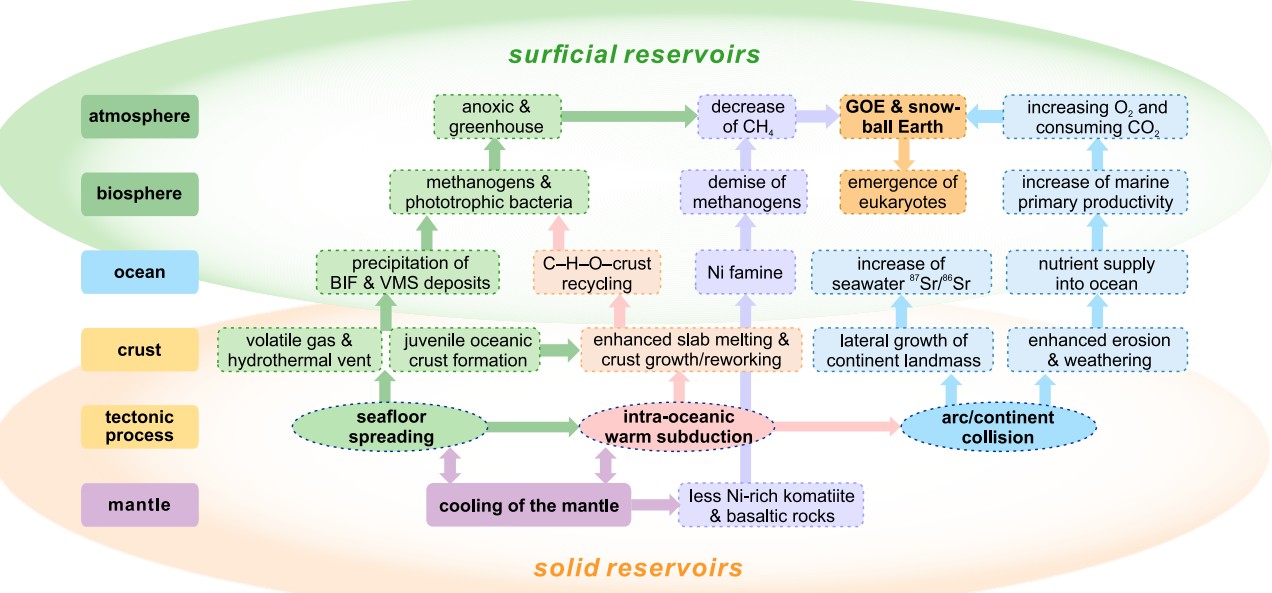

**Fig. 9 | Conceptual model for the operation of widespread plate tectonic processes and its potential influence on surficial environments during the late** **Neoarchean and early Paleoproterozoic.** BIF banded iron formation, GOE Great Oxygenation Event, VMS volcanogenic massive sulfide.

in its near-surface environments, including the ocean, atmosphere, and biosphere (Figs. 8 and 9).

As today, seafloor spreading focused at mid-ocean ridges or in the forearc would have produced large amounts of juvenile oceanic crust that facilitated the heat loss of mantle, the release of volatile species, and intense hydrothermal seafloor alteration. These processes led to an enrichment of Si and key metals (e.g., $Fe^{2+}$, Cu, Zn), and ultimately the precipitation of ca. 2.54–2.51-Ga BIFs and subordinate massive sulfide deposits (VMS) that are common in greenstone belts within the NCC and elsewhere[59]. Meanwhile, the continental shelves outboard of emergent continental landmass, with the development of siliciclastic rocks and/or carbonates and trace elements released from seafloor volcanism and hydrothermal vents, provided hospitable terrestrial ecosystems for phototrophic organisms, which resulted in local "whiffs" of oxygen[16]. The subduction zones that may have extended laterally for >1000 km, as proposed for the Neoarchean central arc system of the NCC, likely promoted the recycling of C–H–O, crust–mantle interaction, crustal growth and reworking during oceanic subduction and arc magmatism. Self-sustaining subduction recycles the C–H–O-enriched oceanic crust into the mantle, generates new crust, and releases volcanic gases through arc magmatism, which were fundamental to regulating Earth's C–H–O cycles and climate.

Around the Archean–Paleoproterozoic boundary, widespread collision between continental fragments led to their lateral growth and the formation of mountain belts[18,50]. Mountain building would have been associated with higher rates of erosion and sedimentation, as evidenced by the thick early Paleoproterozoic (ca. 2.45–2.32 Ga) sedimentary sequence of the lower Songshan Group that is dominated by interlayered quartzite, mica-quartz schist, with minor marble and granular iron formation (GIF) deposited along the Central Orogenic Belt of the NCC[44,60]. Widespread continental emergence and erosion are further supported by a significant increase of seawater $^{87}Sr/^{86}Sr$ in the early Paleoproterozoic (Fig. 8b), which implies enhanced radiogenic Sr input due to weathering of continental crust[61].

Enhanced erosion and weathering of TTG- and greenstone-dominated continental crust would have increased nutrient supply to the oceans, which in turn promoted marine primary productivity, and a significant increase of atmospheric oxygen (GOE) in the early

Paleoproterozoic (ca. 2.4–2.1 Ga, Fig. 8)[16,62], ultimately creating environments habitable to eukaryotes. Furthermore, silicate weathering consumed vast quantities of atmospheric $CO_2$ that, together with the decrease of greenhouse gas $CH_4$ likely induced by the late Neoarchean Ni famine[63], would have contributed to the formation of icehouse conditions, consistent with the ca. 2.4–2.1-Ga Huronian global glaciation[64] (Figs. 8c and 9). The operation of global-scale plate tectonic processes by the late Archean thus likely played one of the fundamental roles in facilitating ready interaction between Earth's interior and surficial environments, a process that has continued until the present day.

## Methods
### Major and trace element analysis
The major and trace elements were analyzed at the State Key Laboratory of Geological Processes and Mineral Resources, China University of Geosciences, Wuhan, China (GPMRCUG) and the Sample Solution Analytical Technology Co., Ltd., Wuhan, China (SSTW). Samples were crushed to a 200-mesh powder in an agate mill. The major oxides were analyzed using an X-ray fluorescence spectrometer (Primus II, Rigaku, Japan). Analytical uncertainties are less than 5%. For trace elements, sample powders were dissolved in a $HNO_3$ + HF solution and measured using an Agilent 7700 inductively coupled plasma-mass spectrometer (ICP-MS). International standards included BHVO-2, BCR-2, and RGM-2. The analytical precision for most trace elements was better than ±5%, except for Li and Be, with relative errors ±10%. The results of major and trace element analyses are presented in Supplementary Data 1.

### Zircon U–Pb and Lu–Hf isotopic analysis
First, cathodoluminescence (CL) images of zircons were obtained by a Gatan Mono CL4 + CL system at the GPMRCUG. Zircon U–Pb isotopic analysis was performed using LA-ICP-MS consisting of a GeoLas HD 193 nm excimer ArF laser ablation system (Coherent Inc., Göttingen, Germany) and an Agilent 7700e ICP-MS at the GPMRCUG. The spot diameter of laser ablation is 32 μm. Zircon 91500 was analyzed twice every 5 analyses as an external standard, and zircon standards GJ-1 and Plešovice were also measured as monitoring standards. All zircon standards yield ages consistent with the recommended values within error

(see Supplementary Data 2). The synthetic silicate glass NIST SRM 610 was used as the standard to calibrate the trace element compositions.

Zircon Lu–Hf isotopic analysis was conducted using LA-MC-ICP-MS consisting of a Neptune Plus MC-ICP-MS (Thermo Fisher Scientific, Germany) in combination with a Geolas HD excimer ArF laser ablation system at the SSTW. All data were acquired on zircon by single spot ablation at a spot size of 44 μm. The spots of zircon Lu–Hf isotopic analysis were over or adjacent to those of zircon U–Pb analysis. Each measurement consisted of 20 s of acquisition of the background signal followed by 50 s of ablation signal acquisition. Zircon 91500 was analyzed twice every 8 analyses as an external standard. Zircon TEM and GJ-1 were used as the monitoring standards. All standards yield $^{176}Hf/^{177}Hf$ values consistent with the recommended reference values (see Supplementary Data 3).

Off-line selection and integration of background and analyte signals, time-drift correction and quantitative calibration of trace element analyses, and data reduction followed the protocols of ref. 65. The zircon U–Pb and Lu–Hf isotopic results are presented in Supplementary Data 2 and 3, respectively.

### Phase equilibrium modeling
Phase equilibrium (pseudosection) diagrams were conducted using THERMOCALC (version 3.45), the internally consistent thermodynamic data set ds62[66] and the $Na_2O–CaO–K_2O–FeO–MgO–Al_2O_3$ $–SiO_2–H_2O–TiO_2–O$ (NCKFMASHTO) chemical system. The a–x of solutions followed ref. 67, including augite (clinopyroxene), garnet, orthopyroxene, hornblende, plagioclase, magnetite–spinel, ilmenite, biotite, white mica, chlorite, and tonalitic melt. The water content was fixed to be just sufficient to saturate the solidus at 1 GPa (producing < 1 mol% $H_2O$-saturated melt). Calculations assume an $Fe^{3+}/\Sigma Fe = 0.1$. The bulk compositions used are presented in Supplementary Data 4.

### Trace element modeling
The composition and abundance of melt and solid (residual) phases at a given pressure and temperature were modeled using the pseudo-section method, which was implemented using the Gibbs free energy minimization program Theriak-Domino[68] with the same conditions as the THERMOCALC calculations, as Theriak-Domino can directly model the mass fractions of different phases. The calculated residual stable phase assemblage ($X$, wt%, mass fraction) and corresponding melt proportion ($F$) were used to calculate bulk partition coefficients [$D_s = \Sigma(K_d \times X)$] from individual published mineral–melt partition coefficients ($K_d$). The partition coefficients (Supplementary Data 5) are from ref. 36. The element compositions of melts can be calculated using the modal batch melting Eq. (1):

$$C_m = \frac{C_o}{F + (1-F)^* D_s} \tag{1}$$

where $C_o$ and $C_m$ are the concentration of elements in the source rock and melt, respectively[69]. To test whether the Angou trondhjemitic gneisses could represent the melt after fractional crystallization, we used the Rayleigh fractional crystallization Eq. (2) to conduct trace element modeling:

$$C_m = \frac{C_o}{F^{(D_s-1)}} \tag{2}$$

The modeled results of modal batch melting and Rayleigh fractional crystallization are presented in Supplementary Data 6 and 7, respectively.

### Data availability
All data used in this study are available in the Supplementary Information/Data files and published literature.

## Code availability
The programs THERMOCALC and Theriak-Domino used in the phase equilibrium calculations are available online (https://hpxeosandthermocalc.org, and https://titan.minpet.unibas.ch/minpet/theriak/theruser.html, respectively).

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

## Acknowledgements

This research was supported by grants from the National Natural Science Foundation of China (NSFC, 42102244 to B.H., 41890834 and 4191144020 to T.K.), the Chinese Ministry of Education (BP0719022), the China Postdoctoral Science Foundation (CPSF, 2021M692977 to B.H.), the Fundamental Research Funds for National Universities (CUG2106365 to B.H.) from the China University of Geosciences, Wuhan, and the MOST Special Fund (MSFGPMR2022-7) from the State Key Laboratory of Geological Processes and Mineral Resources, China University of Geosciences, Wuhan. T.E.J. acknowledges funding from the Australian Government through an Australian Research Council Discovery Project (DP200101104), and support from the State Key Laboratory of Geological Processes and Mineral Resources, China University of Geosciences, Wuhan (Open Fund GPMR202101). A.P. acknowledges NSERC Discovery Grant (RGPIN-2019-04236). D.F. acknowledges support from the NSFC project (42102268) and the CPSF projects (2022T150599 and 2020M682512). We thank Man Liu and Qunye Qian for their assistance with the field investigations.

## Author contributions

B.H. conceived the project and developed the initial idea. B.H. and D.F. carried out field investigations, sampling, and experimental analysis. B.H. undertook the phase equilibria and trace element simulations with suggestions from T.J., and B.H. wrote the original draft. T.K. provided partial funding support for field and sample analysis. B.H., T.J., S.A.W., A.P., D.F., and T.K. contributed to conceptualization, discussion, and revisions.

## Competing interests

The authors declare no competing interests.

## Additional information

**Correspondence and requests** for materials should be addressed to Bo Huang or Timothy Kusky.

