## [Peer Review File · Nature Communications]

REVIEWER COMMENTS

Reviewer #1 (Remarks to the Author):

I have now finished my review of the aforementioned Nature Communications Manuscript which I provide below.

MS DESCRIPTION & SYNOPSIS On the basis of whole rock major and trace element, zircon U-Pb and Lu-Hf isotopes, phase equilibrium modeling and trace element modeling, the authors provide a model stipulating the operation of plate tectonics circa 2.5 Ga.

PRESENTATION & SCIENTIFIC INTERPRETATION The text is very well-written with few syntax or spelling mistakes. The ms will be of broad interest to those working on the origin and timing of plate tectonics on Earth. Main concerns with the manuscript are the omission of key details, data treatment and the tectonic model as I describe below.

MAJOR POINTS:

1. Some omissions: (a) there is no discussion on possible alteration of the lavas studied. 2 However, many of the samples have clearly been altered to high degrees (ones with >3 wt.% LOI). Has the dataset been filtered in any way to omit clearly altered samples from the interpretations? Some samples have >5 wt% LOI and I don't think there is anyway that these can reliably be used. (b) There is not a single mention of mantle rocks in the manuscript (or dikes). More on the mantle rocks below.

2. I don't understand the tectonic model. How is it possible to emplace a MORB unit which is on the downgoing plate? Why would this not be completely subducted? In order for a unit to be emplaced, it needs to be generated in the forearc wedge above the subducting slab and then be subsequently thrust upon failed continent during failed continental subduction. How their idea is accomplished is not shown on Fig. 7. Moreover, studies of the classical Neotethyan forearc-derived and IBM forearc show that the earliest formed subduction initiation (SI) lavas are indeed MORB-like (Whattam and Stern, 2011; Stern et al., 2012; Reagan et al., 2010 and many IBM publications since). Does it not make more sense that the 'MORB' of the Eastern Belt instead represent the earliest formed SI lavas and that the remaining Central and Western Belts were constructed atop? Is this not also supported by nearly overlapping ages? Related to the above, if the tectonic situation was as described in the text, would there not be an associated mantle suite below the subduction lavas separating it from the MORB suite? And in general, I find it odd that there are no mantle rocks but even more so, that is not even mentioned in the text. Though definitions vary on exactly what does or does not constitute an ophiolite, I believe most would agree that it needs to have a mantle component. This needs to at least be addressed in the manuscript. As there is also no mention of dikes, it appears that the suites, if ophiolitic, represent only uppermost carapace.

3. Fig. 1. Is this necessary? If so, a couple of things: (i) the “hallmark” of IOA formation is slabfluid-flux melting of the mantle wedge, but this is even shown on (a). (ii) Also in (a), why is ‘slab-melting’ shown? Perhaps this occurs in some rare instances when there is subduction of hot, young crust (

Reviewer #2 (Remarks to the Author):

The paper by Huang and others is a descriptive study in the Neoproterozoic Angou Complex in the southern North China craton. The paper documents a detailed set of ~2.5 Ga geological units that appear to correspond stratigraphically to modern island arcs later accreted to the eastern block of the China Craton. They represent stages of oceanic crust spreading, basin formation, arc generation and collision. The authors used detailed petrography and petrological descriptions of samples from two distinct terranes combined with U-Pb and Lu-Hf analyses in zircon and modelling to argue that modern-style plate tectonics operated in the Neoproterozoic. I think this is a robust work with detailed field documentation, which is rare in the recent literature that reached similar conclusions. As such, it deserves publication in Nature Communications as long as clarification is provided in key sections of the manuscript. These revisions are important but I believe they can be done quickly.

My two main concerns are with respect to the U-Pb results obtained from zircon grains and the definition of modern-style plate tectonics. It is unclear the temporal extent of ocean spread and basin formation and arc magmatism duration, as there is a substantial overlap at 2.51 Ga. This is particularly caused by the lack of detailed information, even in the supplementary material, about the zircon grains analysed, what are the samples collected in the outcrop pictures and perhaps a diagram that presents a clear temporal number of events to help the reader to understand the geodynamic model proposed. The presence of plate tectonics in the Proterozoic is well supported in the study, but it should not be referred as to modern-style because the thermobaric ratios obtained are too high, and in fact, the authors recognise that the subduction conditions were warmer than what is typical of modern-style subduction.

I think this study would become an excellent contribution to Nature Communications if:

- 1- The use of the word subduction is revised to avoid confusing the reader.
- 2- The U-Pb information of the ages obtained is properly addressed (standards reported, and weighted mean and Concordia plots provided with respective MSWD are presented at least in supplementary material).
- 3- A correlation between field pictures and samples collected is provided.
- 4- Temporal description of events is more clearly presented (how long was the spreading, subduction, arc maturation and collision - estimates for each of these stages in a chart, for example).
- 5- The subducted slab length is recalculated, considering uncertainties in temporal subduction and magmatic events.

Below are the line-by-line comments on the main manuscript PDF file. I hope the authors find

these comments helpful.
Best wishes,
Hugo Moreira, April 2022

Lines 16-17: Yes, I agree with this statement - but please note that the subduction zone may not be characterised by the cold gradients and yet divergent and convergent co-exist. Plate tectonics (rigid lithospheric plates that rearrange themselves across weak boundaries - Cawood et al. 2018) could be present globally in the Archaean, but yet subduction (as we know it today) didn't exist because geothermal conditions were higher, and the plunging oceanic plate did not reach deeper penetration in the mantle. The subducting oceanic plate would break off much shallower than in modern-style plate tectonics. I think a lot of the present debate in the literature could be solved with a better definition of what we call subduction. For example, subcretion (Bedard, 2018 GSF) or hot-subduction (Moyen and Martin, 2012) could be a good compromise for your model. Although the study demonstrates paired metamorphism, it is not as bimodal as one would expect for the Cryogenian.

Lines 45-46: Rephrase. Cold subduction only at ~2.1 Ga. The references proposed here also suggest the same, but the way it is written suggests either that cold subduction occurred earlier or that A-P transition is much later.

Lines 55-57: Modern-style plate tectonics generates pair metamorphic belts whereby high-P/low-T metamorphism occurs in the subduction zone, and low-P/high-T at arc-back-arc as a record of low and high gradients. A primitive form of plate tectonics does not need to have these characteristics. I think it is OK to envisage a mosaic of lithospheric plates by the Archaean, and that might have been global but whether it was modern (like observed today) is a different classification.

Line 99: delete 'ago' then Myr, not Ma.

Line 112: There is also a description of the Easter belt in this section, and perhaps it should be mentioned here? Also, why wasn't sample 18AG08-4-01 included in the results/discussion, but it is present in the supplementary material?

Line 137: The supplementary material should contain the U-Pb Concordia plots of the samples and the spot location of zircon grains ablated. Many zircon grains have core-rim structure, yet there is no information if the ages in Table S2 come from the rims or cores.

Line 150-153: Wouldn't you expect non-mantle like values for the $\delta^{18}O$ values if fluids altered the source? Is this value taken from Zhang et al. 2018 correct? 6.6 ‰ is already beyond mantle values.

Lines 155-157: Adakites also have high Mg, Ni and Cr contents. Is that the case for these rocks of the Central Belt?

Lines 235-237: The issue here is that the associations are coeval, and the ages obtained are not resolvable because they mostly overlap. It would be important to demonstrate that the rocks studied are formed in separate stages, for example, showing the crosscutting field relationship if present.

Lines 256-257: This calculation is fine, but your U-Pb constraints have no such resolution to precisely calculate the slab length. Although it could be 40 Myrs of 'continued subduction', it also could be much shorter given the uncertainties of the ages obtained. These need to be better described in the results section and again in the discussion, otherwise it feels like an assertion. I think the manuscript needs a better description and interpretation of the ages.

Line 268: Plate tectonics, yes, but not modern-style. Calling it 'modern-like' is confusing.

Line 282: I think this is overestimated. How would a >1000 km oceanic plate not start cooling down and potentially generate a lower T/P gradient when subducted?

Line 324: mid not 'mod'.

Line 352: The Huronian glaciation extends to 2.2/2.1 Ga.

Line 366: This section of the paper can be improved. As it stands, it does not give enough detail for a proper assessment of the quality of data. Please, provide the analysed standards for U-Pb or Lu-Hf analyses.

Supplementary material -

Fig. S9: Caption says the weighted age is provided, but I can't see it anywhere. Could you also add the U-Pb and Lu-Hf spot location of analysed grains?

Reviewer #3 (Remarks to the Author):

To decipher the Archean plate tectonics activated prior to the present-day North China Craton, the authors divided the Angou Complex, southern Paleoproterozoic Trans-North China Orogen into three litho-structural belts: (1) the Eastern Belt, comprising metabasalt and fine-grained metasedimentary sequences, being regarded to be formed in the Archean spreading oceanic crust; (2) the Central Belt, composed of metamorphosed mafic to felsic volcanic rocks and sedimentary successions, ascribed to island arc basalt and related rocks; and (3) the Western Belt, consisting mainly of TTG gneiss and minor metabasite with the later being ascribed to MORB-like basalt. Age data indicate that tectonic juxtaposition of these three different belts occurred at or before 2.50 Ga, during arc-continent collisional orogenesis. The authors suggest that the Eastern Belt of the Angou Complex is one of the oldest passive margin sequences, whereas the Central and Western belts represent a SSZ arc/forearc setting. Combined with literature data, the authors rebuilt the late Archean plate tectonics of the "Proto-North China Ocean". The results and inference are quite interesting.

However, some questions should be considered and they are listed as follows.

(a) Although the authors derived that the “Eastern Belt” of the Angou Complex formed at a spreading ridge outboard of a passive margin, but how could the BIF be formed at such tectonic setting? BIF is repeatedly certified to be formed at continental margin or shallow sea or lake in the Archean or Paleoproterozoic. BIF also appears in the “Central Belt”.

(b) The “Western Belt” of the Angou Complex consists mainly of TTG gneiss and minor amphibolite, which signify continental rather than oceanic origin.

(c) All the rocks were metamorphosed at different extent. Therefore, the authors are suggested to specify what geochemical data (e.g., immobile elements) can be used to discriminate the tectonic settings of these rocks, in order to yield unambiguous results.

(d) About thermodynamic modeling. Because the authors used averaged MORB (MORB-AV) and IAB (IBA-AV) compositions as starting materials, the results should be ubiquitous to all such rocks. Please specify this. Furthermore, it seems there are similar modeling work already reported in the literature. So, please describe what are the differences between the modeling results of this manuscript and others’?

Specific comments:

Abstract: The study area should be given clearly.

Lines 40 and 42. “Earth” can be revised to be “Earth’s”.

Lines 66, 258 and 259. Please check and revise the term “~1600 km-long.....”. In fact, the Trans-North China Orogen is no more than 500 km long.

Line 82. “2.50 Ga” can be corrected as “~2.45 Ga”.

Lines 125-127. To decide the maximum depositional age of a sedimentary sequence, only the youngest age of detrital zircons can be used. The authors should explain why 2533±13 Ma was estimated to be the maximum depositional age.

Line 167. “is” is suggested to corrected as “are”.

Line 165. About the “Thermodynamic and trace element modelling” section. The author may cite direct petrological experiments if there are such data, rather than just on numerical modeling results.

Line 317. About the “Implications for the Neoproterozoic surficial environments”. This section does not match the topic of this manuscript and can be deleted.

Line 324. “mod-ocean” should be corrected as “mid-ocean”.

The reference list: The cited journals are in either full names or abbreviations, so the style should be unified.

Line 413. Authors are lost.

Line 499. “Lithosphere-US.” Should be corrected as “Lithosphere”.

Line 547. “advances” should be corrected as “Advances”.

Figure 1 is unnecessary but can be deposited as an electronic appendix.

Figure 7a. What are the pink regions?

Figure 8. Routine description of thermal gradient is in °C/kbar or °C/km rather than in °C/GPa.

Response Letter to Reviewers

Dear reviewers,

We are thankful for the thorough reviews, and constructive comments/suggestions, which have been very helpful for us to improve the quality of the paper. We appreciate these comments and suggestions, and implemented careful revisions and addressed all concerns accordingly. The detailed revisions can be found in the “Manuscript with changes tracked (in red)”, and the point-by-point response as follows (reply in blue).

POINT-BY-POINT RESPONSE TO REVIEWER COMMENTS (reply in blue)

Reviewer #1 (Scott A. Whattam):

MS DESCRIPTION & SYNOPSIS

On the basis of whole rock major and trace element, zircon U-Pb and Lu-Hf isotopes, phase equilibrium modeling and trace element modeling, the authors provide a model stipulating the operation of plate tectonics circa 2.5 Ga.

PRESENTATION & SCIENTIFIC INTERPRETATION

The text is very well-written with few syntax or spelling mistakes. The ms will be of broad interest to those working on the origin and timing of plate tectonics on Earth. Main concerns with the manuscript are the omission of key details, data treatment and the tectonic model as I describe below.

We appreciate Prof. Scott A. Whattam for the positive recommendation and constructive comments. We have implemented relevant revisions and addressed all concerns accordingly.

MAJOR POINTS:

1. Some omissions: (a) there is no discussion on possible alteration of the lavas studied. However, many of the samples have clearly been altered to high degrees (ones with >3 wt.% LOI). Has the dataset been filtered in any way to omit clearly altered samples from the interpretations? Some samples have >5 wt% LOI and I don't think there is anyway that these can reliably be used. (b) There is not a single mention of mantle rocks in the manuscript (or dikes). More on the mantle rocks below.

Yes, thanks for these important comments. We have implemented relevant revisions as outlined below.

(a) In the revised version, we excluded the highly-altered samples (LOI >5 wt%, n=5) and mainly used immobile elements in the discussion. We have added a detailed discussion on elemental mobility (for details please see Supporting Information), which ensures that the data and proxies used in the discussion are free from the significant influence of alteration or metamorphism.

(b) Ophiolite refers to on-land fragments of the oceanic lithosphere. An ideal section of ophiolites includes, from base up, ultramafic mantle tectonites, gabbroic rocks, diabase sheeted dikes, and basaltic lavas and overlying chert-shale. But the structures, components, and origins of natural examples of on-land ophiolites or present-day oceanic lithosphere are much more complex, with many components (e.g., sheeted dikes) absent in most cases. In ancient orogenic belts, the dismembered ophiolites may also lack significant ultramafic components, which were attributed to a lack of exhumation/accretion during subduction or post-obduction/collision erosion.

Ultramafic mantle rocks are absent from the Angou Complex, but ultramafic components (e.g., pyroxenite) occur in the adjacent Dengfeng Complex. More significant volumes of ultramafic rocks, including chromitites, also occur in coeval greenstone belts in the central orogen of the North China Craton (e.g., adjacent Wutaishan, Hongqiyingzi, and Zunhua complexes, e.g., Kusky et al., 2001, 2022; Polat et al., 2005, 2006).

In this work, we didn't use the term of "ophiolite". See below for more detailed explanations on the "ophiolite", mantle rocks, and mafic dikes.

Kusky, T. M., Li, J. H. & Tucker, R. D., The Archean Dongwanzi ophiolite complex, North China craton: 2.505-billion-year-old oceanic crust and mantle. *Science*. **292**, 1142-1145 (2001).

Polat, A., Herzberg, C., Munker, C., Rodgers, R., Kusky, T., Li, J., Fryer, B. & Delaney, J., Geochemical and petrological evidence for a suprasubduction zone origin of Neoproterozoic (ca. 2.5 Ga) peridotites, central orogenic belt, North China craton. *Geol. Soc. Am. Bull.* **118**, 771-784 (2006).

Polat, A., Kusky, T., Li, J.H., Fryer, B., Kerrich, R. & Patrick, K., Geochemistry of Neoproterozoic (ca. 2.55-2.50 Ga) volcanic and ophiolitic rocks in the Wutaishan greenstone belt, central orogenic belt, North China craton: Implications for geodynamic setting and continental growth. *Geol. Soc. Am. Bull.* **117**, 1387-1399 (2005).

Kusky, T., Wang, J., Wang, L., Huang, B., Ning, W., Fu, D., Peng, H., Deng, H., Polat, A., Zhong, Y. & Shi, G., Mélanges through time: Life cycle of the world's largest Archean mélange compared with Mesozoic and Paleozoic subduction-accretion-collision mélanges. *Earth-Sci. Rev.* **209**, 103303 (2020).

2. I don't understand the tectonic model. How is it possible to emplace a MORB unit which is on the downgoing plate? Why would this not be completely subducted? In order for a unit to be emplaced, it needs to be generated in the forearc wedge above the subducting slab and then be subsequently thrust upon failed continent during failed continental subduction. How their idea is accomplished is not shown on Fig. 7. Moreover, studies of the classical Neotethyan forearc-derived and IBM forearc show that the earliest formed subduction initiation (SI) lavas are indeed MORB-like (Whattam and Stern, 2011; Stern et al., 2012; Reagan et al., 2010 and many IBM publications since). Does it not make more sense that the 'MORB' of the Eastern Belt instead represent the earliest formed SI lavas and that the remaining Central and Western Belts were constructed atop? Is this not also supported by nearly overlapping ages?

Thanks for these insightful comments and helpful suggestions. Indeed, most of the Cretaceous (Neotethyan) huge ophiolite sheets formed in forearcs, and were emplaced (obducted) on the facing continental margins. Really spectacular, and well described by the reviewer, and many others (e.g., Bob Stern) (e.g., Whattam and Stern, 2011; Stern et al., 2012). This type of ophiolite formation is one case, however, and not the only way to form oceanic crustal remnants within accretionary systems in orogens. With due respect, we agree with this formation and obduction emplacement mechanism, but also, from experience, suggest that the reviewer considers the other cases (subduction-accretion), well documented. A further point is that these huge Neotethyan ophiolites are sitting in the uppermost crust levels, and do not extend deeply. When the Tethyan collision is finally over, they will be gone, and only small remnants left, like those we describe here. See detailed point-by-point response below.

Whattam, S. A. & Stern, R. J. The 'subduction initiation rule': a key for linking ophiolites, intra-oceanic forearcs, and subduction initiation. *Contrib. Mineral. Petr.* **162**, 1031-1045 (2011).

Stern, R. J., Reagan, M., Ishizuka, O., Ohara, Y. & Whattam, S. To understand subduction initiation, study forearc crust: To understand forearc crust, study ophiolites. *Lithosphere.* **4**, 469-483 (2012).

Related to the above, if the tectonic situation was as described in the text, would there not be an associated mantle suite below the subduction lavas separating it from the MORB

suite? And in general, I find it odd that there are no mantle rocks but even more so, that is not even mentioned in the text. Though definitions vary on exactly what does or does not constitute an ophiolite, I believe most would agree that it needs to have a mantle component. This needs to at least be addressed in the manuscript. As there is also no mention of dikes, it appears that the suites, if ophiolitic, represent only uppermost carapace.

We addressed these concerns as detailed below:

1) Emplacement processes of the MORB unit.

As the reviewer will be aware, the emplacement of oceanic lithosphere as “ophiolites” is complicated, but can be subdivided into two main processes: obduction, and subduction-accretion. Note also that we do not use the term ophiolite considering (i) we do not have a “full” ophiolite suite as in the huge Tethyan ophiolite massifs the reviewer mentions, but only a dismembered or partial sequence using the 1972 Penrose definition, or ophiirags using the Sengor and Natal'in 2004 definition, or just slivers of uppermost ocean crust and overlying sediments, called OPS or ocean plate stratigraphy in the Kusky et al. 2013 definition. (ii) this term may be controversial for Precambrian greenstone belts; we just use the more objective “island arc”, “forearc”, “mid-ocean ridge” to describe the lithological assemblage (e.g., forearc complex) and inferred tectonic settings.

A model for subduction and accretion of ocean plate stratigraphy (Kusky et al., 2013).

Obduction of SSZ-type forearc/island arc complexes generally forms relatively intact ophiolitic sheets, such as those in Oman and Troodos, which do include mantle components. By contrast, accretion of MOR- or OIB-type rocks derived from the subducting slab leads to formation of structurally-dismembered accretionary complexes that can contain relatively coherent ocean plate stratigraphy units (basalt-chert ± shale ± limestone ± trench turbidites) and chaotic block-in-matrix mélangé units (e.g., Wakabayashi, 2017), but which may lack mantle components, probably due to their high density.

Obduction generally occurs during arc-continental margin or arc-arc collision during subduction termination, whereas subduction-accretion can occur while subduction is ongoing (when the subduction margin is accretionary rather than erosional). The mechanisms of accretion include offscraping, underplating along decollements, diapirism, etc. The accreted oceanic rocks (accretionary ophiolites/ocean plate stratigraphy) from downgoing plates can range from centimeters to several kilometers thick, and is a function of subduction angle and velocity, the thickness of sediments atop the oceanic crust, and the depth of the decollement (e.g., Kusky et al., 2020, Earth Sic. Rev.). For example, accretionary complexes in circum-Pacific accretionary orogens, such as those in Japan (e.g., Wakita, 2005), Alaska (e.g., Kusky and Bradley, 1999; Kusky et al., 1997), and Coast Ranges in the USA (e.g., Wakabayashi, 2017) can reach several kilometers in thickness.

A model for accretion of ocean plate stratigraphy (accretionary complex) in Alaska (Kusky et al., 2020, ESR).

The magmatic components of accreted ophiolites commonly have an ocean plate stratigraphy and are dominated by MORB and OIB, with subordinate SSZ components derived from olistostrome or subduction erosion of the upper plate. In our case, in the adjacent Dengfeng Complex, we have identified an accreted lower-plate assemblage including lower-plate ocean plate stratigraphy, block-in-matrix mélanges and accreted trench-fill turbidites, and a relatively intact upper-plate arc/forearc assemblage (Huang et al., 2019, *GSA Bulletin*; 2020, *EPSL*). In the Angou Complex, the MORB-like unit in the Eastern Belt contains lithological associations of MOR-type ocean plate stratigraphy and is in thrust contact with an inferred upper-plate unit (Western and Central belts), which was most likely accreted to the forearc accretionary wedge. In summary, we think it is reasonable to interpret the MORB unit as having been derived from the lower plate and emplaced in the Angou Complex. We have added an explanation of the accretion of MORB-like unit (Lines 295–296, 335–338, Fig. S1a).

2) Tectonic affinity of the MORB-like unit.

In the context of plate tectonics, MORB-like rocks can be generated at mid-ocean ridges, backarc spreading ridges, and forearc spreading ridges (the latter known as forearc basalt, FAB; or proto-arc basalt). Geochemically, they are very similar, with MORB-like REE patterns with or without minor contributions from subduction-related components.

In the Angou Complex, we consider that the MORB unit in the Eastern Belt was most likely generated at a MOR and emplaced during subduction-accretion due to the following evidence:

- (i) **geochronological:** According to the “subduction initiation rule” made popular by Whattam and Stern, (2011), the forearc basalt is the earliest magmatism generated by the decompression melting of upwelling asthenospheric mantle. The MORB unit (~2.53–2.51 Ga) is younger than the FAB (~2.55 Ga) and IAB (~2.54–2.52 Ga) units in the Angou and Dengfeng complexes (Fig. S13). This argues that the oldest MORB-like unit (2.55 Ga) from the Western Belt, instead of the younger Eastern Belt, accords with the forearc subduction initiation model.
- (ii) **geochemical:** the basalts from the Eastern Belt possess MORB-like rather than subduction-related compositions (Figs. 2c–d and 3b–c).
- (iii) **stratigraphic:** the MORB-like basalt–chert–shale–BIF sequences in the Eastern belt are typical lithological associations of MOR-type ocean plate stratigraphy, whereas

forearc basalts–arc volcanic-plutonic rocks in the Western and Central belts are consistent with a typical forearc to arc assemblage.

(iv) structural: the MORB-like unit in the Eastern belt is characterised by repeated ocean plate stratigraphic sequences, which were more likely accreted to the western arc/forearc margin during subduction-accretion rather than obducted forearc sequences that are generally intact in structure.

We have added these to the discussion section (Lines 241–252).

3) Mantle rocks and dikes.

As before, ideal ophiolite sequences involve mantle rocks and sheeted dikes. However, many Phanerozoic ophiolites don't contain these two components, due to the lack of exhumation or post-emplacement erosion. The general lack of ultramafic rocks and sheeted dikes in Archean greenstone belts may be attributed to the difficulty of exhumation due to:

(i) Archean oceanic crust was likely much thicker than its modern counterpart, such that the dense lower-level ultramafic mantle tectonites would have been even more difficult to exhume compared with the upper-level oceanic crust. This is true in many Phanerozoic orogenic belts, such as the Japan accretionary orogen where the accreted rocks are dominated by basalts and overlying sediments (ocean plate stratigraphy) that form the higher levels of oceanic plate. That does not mean the ultramafic portions did not exist. For example, further north in the Central orogenic belt of the NCC, minor sheets or blocks of Neoproterozoic SSZ-type ultramafic rocks that are associated with high-Cr chromitites are exposed in the Wutaishan, Hongqiyingzi, and Zunhua complexes (e.g., Kusky et al., 2020; Polat et al., 2005, 2006).

(ii) Orogenic belts that experience collision have less potential to preserve ultramafic components due to strong erosion and dismemberment. We can observe many excellent on-land ophiolites in Oman, Troodos, and the islands in the Western Pacific Ocean, which may not be preserved as we observed today if they undergo collisional orogenesis millions of years later.

Mélanges in the Neoproterozoic central orogen of the North China Craton (Kusky et al., 2020, ESR). The mélanges in the central and northern segments (e.g., WT–Wutai, EH–Zunhua, Eastern Hebei; JP–Jianping) contain ultramafic rocks.

With respect to the diabase dikes, whereas we do see mafic dikes intruding basaltic lavas in the Angou Complex (Fig. S4g), whether or not they are sheeted dykes is hard to say. The formation of sheeted dykes may be related to the balance between magma supply and spreading rate (e.g., Dilek and Furnes, 2011, GSA Bulletin; Kusky et al., 2011, Sci China Earth Sci). If the spreading rate is fast with insufficient magma supply, the sheeted dykes would be expected to be injected only into extensional rift/fracture zones. However, it's notable that about 90% of ophiolites in the world lack such sheeted dikes (Robinson and Dilek, 2008, GSA Today). In our case, minor diabase dykes intruding the MORB-type metabasalts may suggest sufficient magma supply and a moderate spreading rate, but it is hard to be conclusive. We thus didn't emphasize this point, and just provide information on mafic dikes (Fig. S4).

Considering all of the field, structural, geochronological, petrological and geochemical data above, we therefore argue that our current tectonic model best and simply explains the field, geochemical and structural data, and it fits better with the broader Neoproterozoic tectonic evolution of the central orogen of the NCC.

Dilek, Y. & Furnes, H. Ophiolite genesis and global tectonics: Geochemical and tectonic fingerprinting of ancient oceanic lithosphere. *Geol. Soc. Am. Bull.* **123**, 387-411 (2011).

Kusky, T. M., Bradley, D. C., Haeussler, P. J. & Karl, S. Controls on accretion of flysch and mélangé belts at

- convergent margins: Evidence from the Chugach Bay thrust and Iceworm mélangé, Chugach accretionary wedge, Alaska. *Tectonics*. **16**, 855-878 (1997).
- Kusky, T. M. & Bradley, D. C. Kinematic analysis of mélangé fabrics: examples and applications from the McHugh Complex, Kenai Peninsula, Alaska. *Journal of Structural Geology*. **21**, 1773-1796 (1999).
- Kusky, T. M., Windley, B.F., Safonova, I., Wakita, K., Wakabayashi, J., Polat, A., Santosh, M., Recognition of ocean plate stratigraphy in accretionary orogens through Earth history: A record of 3.8 billion years of sea floor spreading, subduction, and accretion. *Gondwana Res.* **24**, 501-547 (2013).
- Polat, A. et al. Geochemistry of Neoproterozoic (ca. 2.55-2.50 Ga) volcanic and ophiolitic rocks in the Wutaishan greenstone belt, central orogenic belt, North China craton: Implications for geodynamic setting and continental growth. *Geol. Soc. Am. Bull.* **117**, 1387-1399 (2005).
- Polat, A. et al. Geochemical and petrological evidence for a suprasubduction zone origin of Neoproterozoic (ca. 2.5 Ga) peridotites, central orogenic belt, North China craton. *Geol. Soc. Am. Bull.* **118**, 771-784 (2006).
- Robinson, P. T., Malpas, J., Dilek, Y. & Zhou, M. The significance of sheeted dike complexes in ophiolites. *GSA Today*. **18**, 4 (2008).
- Şengör, A. & Natal'in, B. A. Phanerozoic analogues of Archaean oceanic basement fragments: Altaid ophiolites and ophiirags. *Developments in Precambrian Geology*. **13**, 675-726 (2004).
- Wakabayashi, J. Structural context and variation of ocean plate stratigraphy, Franciscan Complex, California: insight into mélangé origins and subduction-accretion processes. *Progress in Earth and Planetary Science*. **4**, 10-1186 (2017).
- Wakita, K. & Metcalfe, I. Ocean Plate Stratigraphy in East and Southeast Asia. *J. Asian Earth Sci.* **24**, 679-702 (2005).
- Whattam, S. A. & Stern, R. J. The 'subduction initiation rule': a key for linking ophiolites, intra-oceanic forearcs, and subduction initiation. *Contrib. Mineral. Petr.* **162**, 1031-1045 (2011).

3. Fig. 1. Is this necessary? If so, a couple of things: (i) the “hallmark” of IOA formation is slab fluid-flux melting of the mantle wedge, but this is even shown on (a). (ii) Also in (a), why is ‘slab-melting’ shown? Perhaps this occurs in some rare instances when there is subduction of hot, young crust (<20 m.y.) but otherwise not ‘typical’ of arc subduction.

Nice comments, we agree, and have now moved this figure into the supporting information after the following improvements:

- (i) We added the slab-fluid-fluxed melting of the mantle wedge;
- (ii) We deleted “slab melting” in the figure, and added a comment in the figure caption for “slab melting”, which only occurs along warm subduction zone geotherms due to ridge subduction, subduction of a young and hot slab, or hotter mantle.

We thank and acknowledge Prof. Whattam for such extensive constructive comments, which we considered carefully and implemented to the revised version of the text as noted.

=====

Reviewer #2 (Hugo Moreira):

The paper by Huang and others is a descriptive study in the Neoproterozoic Angou Complex in the southern North China craton. The paper documents a detailed set of ~2.5 Ga geological units that appear to correspond stratigraphically to modern island arcs later accreted to the eastern block of the China Craton. They represent stages of oceanic crust spreading, basin formation, arc generation and collision. The authors used detailed petrography and petrological descriptions of samples from two distinct terranes combined with U-Pb and Lu-Hf analyses in zircon and modelling to argue that modern-style plate tectonics operated in the Neoproterozoic. I think this is a robust work with detailed field documentation, which is rare in the recent literature that reached similar conclusions. As such, it deserves publication in Nature Communications as long as clarification is provided in key sections of the manuscript. These revisions are important but I believe they can be done quickly.

My two main concerns are with respect to the U-Pb results obtained from zircon grains and the definition of modern-style plate tectonics. It is unclear the temporal extent of ocean spread and basin formation and arc magmatism duration, as there is a substantial overlap at 2.51 Ga. This is particularly caused by the lack of detailed information, even in the supplementary material, about the zircon grains analysed, what are the samples collected in the outcrop pictures and perhaps a diagram that presents a clear temporal number of events to help the reader to understand the geodynamic model proposed. The presence of plate tectonics in the Proterozoic is well supported in the study, but it should not be referred as to modern-style because the thermobaric ratios obtained are too high, and in fact, the authors recognise that the subduction conditions were warmer than what is typical of modern-style subduction.

We appreciate Dr Hugo Moreira very much for the positive recommendation and constructive comments. We agree with all these concerns or suggestions, and conducted appropriate revisions and gave a detailed response below:

I think this study would become an excellent contribution to Nature Communications if:

1- The use of the word subduction is revised to avoid confusing the reader.

Good comment, we agree. We deleted the relevant expression of “modern-like” plate

tectonics to avoid confusion. Based on our calculation of subduction zone geotherms, we propose that the island arc magmas were generated at a warm (or hot) and shallow subduction zone, similar to the present-day “hot” subduction zone as the reviewer suggested (hot-subduction model, Moyen and Martin, 2012). We have also added a sentence to describe the change of subduction style in the discussion (**Lines 351–355**).

2- The U-Pb information of the ages obtained is properly addressed (standards reported, and weighted mean and Concordia plots provided with respective MSWD are presented at least in supplementary material).

Fair point. We have thoroughly improved the presentation of the zircon U-Pb dating, including:

- (i) updated CL images with the analysed spots marked, and added zircon concordia diagrams including MSWD and analytical spot numbers (**Figs. S10 & S11**).
- (ii) added more information on the standards both in the text and supplementary table.
- (iii) provided a detailed description of the results of zircon U-Pb dating and Lu-Hf isotopes in the supplementary text.

3- A correlation between field pictures and samples collected is provided.

We have now marked the sample numbers in the supplementary field pictures (**Fig. S9**) and GPS locations and added sample localities to **Fig. 1b**.

4- Temporal description of events is more clearly presented (how long was the spreading, subduction, arc maturation and collision - estimates for each of these stages in a chart, for example).

Good point. We have added a summary table to show all available ages from the Angou and Dengfeng complexes, and inserted a chart (**Fig. S13**) to show the timing of each process (see below).

Fig. S13 New chart showing ages of rocks and the timing of different events.

5- The subducted slab length is recalculated, considering uncertainties in temporal subduction and magmatic events.

Thanks for this good comment. As the reviewer will be aware, the calculation of subducted slab length is complicated with several possible scenarios that change in terms of the location of subduction initiation, spreading and subducting rate, and duration of subduction. Here, we just presented a simplified and modest estimate using a modest absolute plate velocity between the Central arc and the Eastern Block of the NCC (don't consider the location of subduction and spreading and subduction rates). The key in the simplified estimate is the duration of subduction.

The Neoproterozoic subduction events with continuous arc magmatism from 2.56/2.55 to 2.51/2.50 Ga (with minor spatial variation in the ~1600-km-long Central Orogenic Belt) have been well constrained in the North China Craton. The reported oldest magmatic rocks in this episode of subduction include 2549 ± 9 Ma (forearc metabasalt, this study) and 2554 ± 10 Ma/ 2551 ± 10 Ma (forearc metagabbro, recalculated from Deng et al., 2022) in the Dengfeng and Angou complexes (Fig. S13), and 2566 ± 13 Ma and 2555 ± 6 Ma (SHRIMP zircon age) TTG gneisses in the Wutai Complex (Wilde et al., 2005). The youngest magmatic rocks related to this subduction have been suggested to be at 2.51 Ga or 2.50 Ga (e.g., the youngest 2502–2506-Ga high-Mg diorite in the Dengfeng Complex). Thus, the duration of subduction of 40 Ma in the Central Orogenic Belt is a modest and reasonable

estimate.

If there was an open ocean and a continent at 2.55 Ga, and the ocean was closed at 2.51 Ga

In this assumed case, 3000-km-long slab can be subducted.

Cartoon showing two assumed subduction models with different locations of subduction initiation, and subduction/spreading velocity. The modest estimate of subducted oceanic slab is about 1000 km.

In a simplest model (Example 2, see the cartoon above), assuming a conservative convergence rate of 25 km/Myr, a duration of subduction of ~40 Myr would lead to subduction of >1000 km of oceanic slab. If considering an additional uncertainty on the duration of subduction of ± 10 Ma gives an uncertainty of ± 250 km. If higher subduction velocity is considered (it would be possible in the Archean Earth, when mantle was hotter and mantle convection and subduction may have been faster), then a much longer slab would have been subducted. Such voluminous, hydrous subducted slab provided necessary source rocks for slab melting (melting degree ~7–15 wt. % calculated here) and generating voluminous TTG and adakitic arc rocks.

Wilde, S. A., Cawood, P. A., Wang, K. & Nemchin, A. A. Granitoid evolution in the Late Archean Wutai Complex, North China Craton. *J. Asian Earth Sci.* **24**, 597-613 (2005).

Below are the line-by-line comments on the main manuscript PDF file. I hope the authors find these comments helpful.

Best wishes,

Hugo Moreira, April 2022

We sincerely appreciate Dr Hugo Moreira for such insightful and constructive comments and suggestions. We made a detailed revisions and responses as noted.

Lines 16-17: Yes, I agree with this statement - but please note that the subduction zone may not be characterised by the cold gradients and yet divergent and convergent co-exist. Plate tectonics (rigid lithospheric plates that rearrange themselves across weak boundaries - Cawood et al. 2018) could be present globally in the Archaean, but yet subduction (as we know it today) didn't exist because geothermal conditions were higher, and the plunging oceanic plate did not reach deeper penetration in the mantle. The subducting oceanic plate would break off much shallower than in modern-style plate tectonics. I think a lot of the present debate in the literature could be solved with a better definition of what we call subduction. For example, subcretion (Bedard, 2018 GSF) or hot-subduction (Moyen and Martin, 2012) could be a good compromise for your model. Although the study demonstrates paired metamorphism, it is not as bimodal as one would expect for the Cryogenian.

Good points. We agree that, even in present-day Earth, subduction zones have diverse characteristics, for example, cold vs. warm geotherms, steep vs. shallow subduction angles, fast vs. slow subduction velocity, etc... If cold and deep subduction, with these varying features, is required for defining the “modern-style” plate tectonics (subduction), then this at least locally started at circa 2.2/2.1–2.0 Ga when HP-LT metamorphic rocks are found in localities worldwide, likely associated with the assembly of Nuna/Columbia (e.g., Brown et al., 2020). Widespread paired (bimodal) metamorphism also appeared during this stage (e.g., Brown et al., 2020; Holder et al., 2019 Nature). Based on our modelling work, we propose that the island arc magmas were generated at a warm (or hot) and shallow subduction zone, similar to the “hot” subduction model (Moyen and Martin, 2012). Accordingly, we added a sentence in the discussion section to clarify this change in the style of plate tectonics (subduction) (Lines 351–355).

Holder, R. M., Viète, D. R., Brown, M. & Johnson, T. E. Metamorphism and the evolution of plate tectonics. *Nature*. **572**, 378-381 (2019).
Moyen, J. & Martin, H. Forty years of TTG research. *Lithos*. **148**, 312-336 (2012).

Lines 45-46: Rephrase. Cold subduction only at ~2.1 Ga. The references proposed here also suggest the same, but the way it is written suggests either that cold subduction occurred earlier or that A-P transition is much later.

Rephrased.

Lines 55-57: Modern-style plate tectonics generates pair metamorphic belts whereby high-P/low-T metamorphism occurs in the subduction zone, and low-P/high-T at arc-back-arc as a record of low and high gradients. A primitive form of plate tectonics does not need to have these characteristics. I think it is OK to envisage a mosaic of lithospheric plates by the Archaean, and that might have been global but whether it was modern (like observed today) is a different classification.

Yes, we agree that the style in the Neoproterozoic was becoming more similar to the “modern style”, albeit with warmer subduction zones that enabled intense slab melting and probably more frequent and shallower slab break-off. We rephrased this.

Line 99: delete 'ago' then Myr, not Ma.

Here we describe the precise age of a rock, so we use Ma that means millions of years (Myr) ago. We use Myr when it is a time-gap that we are explaining (for example, the timing of subduction duration ~40 Myr).

Line 112: There is also a description of the Easter belt in this section, and perhaps it should be mentioned here? Also, why wasn't sample 18AG08-4-01 included in the results/discussion, but it is present in the supplementary material?

Thanks for this suggestion. (i) We mainly focused on describing the Central and Western belts in this section, so we didn't include it in the subtitle of this section. (ii) Sample 18AG08-4 is a K-rich granite dike intruding the volcano-sedimentary unit of the Angou Complex (Fig. 1b and Fig. S6e). We added the relevant information of sample 18AG08-4, which provides a constraint on the age of tectonic stacking.

Line 137: The supplementary material should contain the U-Pb Concordia plots of the samples and the spot location of zircon grains ablated. Many zircon grains have core-rim structure, yet there is no information if the ages in Table S2 come from the rims or cores.

Thanks. We have updated the materials related to zircon U-Pb dating, including the addition of U-Pb concordia diagrams (Fig. S11), and the analysed spot location of representative zircon grains on CL images (Fig. S10). Also, we carefully checked the zircon texture, and most analysed zircons are free of core-rim structure, except for zircons from an amphibolite sample (22RZ06b), as shown in the updated CL images (Fig. S10). We have also added a detailed description of zircon U-Pb dating results in supporting information.

Line 150-153: Wouldn't you expect non-mantle like values for the d18O values if fluids altered the source? Is this value taken from Zhang et al. 2018 correct? 6.6 ‰ is already beyond mantle values.

Thanks for pointing out this. One metabasalt sample in Zhang et al. (2018) has above-mantle zircon d18O values, which have been attributed to the contribution of slab-derived fluids. We now focused on our data and deleted this citation here for not to cause debate or uncertainty.

Lines 155-157: Adakites also have high Mg, Ni and Cr contents. Is that the case for these rocks of the Central Belt?

Good comment. One of the features of adakites is the relative enrichment of Mg, Cr and Ni, which has been suggested to reflect the equilibration interaction between slab melt and mantle peridotite. However, this feature is absent in many Phanerozoic adakite samples, probably due to the fast ascent of slab melts (not enough time to reach melt equilibration interaction), or shallow subduction angles (with little space to interact with mantle wedge) (e.g., Smithies et al., 2003; Castillo, 2012). Some samples (e.g., 22AG43, 18AG17-6, 18AG05-4) in the felsic volcanic rocks of the Central Belt have relatively high Mg#, MgO (1.61–2.97 wt%), Cr (32–96.5 ppm), and Ni (24–34.7 ppm) contents (Table S1), consistent with possible interaction with the mantle wedge. Note that the space of the mantle wedge may have been changed during continuous subduction and slab roll-back. In addition, locally, high-Mg basalt and diorite were generated by partial melting of the mantle wedge, and the latter (Si-, Mg-enriched dioritic magma) possibly involved interaction with slab-

derived high-Si melts. Thus, the overall suite of igneous rocks in the Central Belt is comparable with “hot-subduction” magmatic suites (Moyen and Martin, 2012).

Castillo, P.R., Adakite petrogenesis. *Lithos* **134-135**, 304-316 (2012).

Smithies, R.H., Champion, D.C. & Cassidy, K.F., Formation of Earth's early Archaean continental crust. *Precambrian Res.* **127** (1-3), 89– 101 (2003).

Moyen, J. & Martin, H. Forty years of TTG research. *Lithos.* **148**, 312-336 (2012).

Lines 235-237: The issue here is that the associations are coeval, and the ages obtained are not resolvable because they mostly overlap. It would be important to demonstrate that the rocks studied are formed in separate stages, for example, showing the crosscutting field relationship if present.

Thanks for this good comment. We can constrain this by using the following considerations: 1) as replied before, we collected all available geochronological data (zircon U-Pb dating, Table S9), and made a geochronological sequence chart (Fig. S13), which shows the age relations between different lithological units. The oldest ages are from the MORB-like metabasites (forearc, ca. 2.55–2.54 Ga), and then arc-affinity TTG and volcanic rocks (ca. 2.54-2.51 Ga), and high-Mg diorites (ca. 2.53–2.50 Ga). All of these units were intruded by post-kinematic granitic and mafic dikes (ca. 2.51-2.4 Ga). 2) The different mantle source characteristics (unmodified MORB-like FAB vs. enriched IAB) reflected by the basaltic rocks in the Western and Central belts are consistent with a temporal progression of magmatic sequences (e.g., subduction initiation to arc maturation) during continuous subduction, rather than differences in the distance to the subduction zone (trench-distal vs. trench proximal). Thus, we consider our interpretation reasonable.

Lines 256-257: This calculation is fine, but your U-Pb constraints have no such resolution to precisely calculate the slab length. Although it could be 40 Myrs of 'continued subduction', it also could be much shorter given the uncertainties of the ages obtained. These need to be better described in the results section and again in the discussion, otherwise it feels like an assertion. I think the manuscript needs a better description and interpretation of the ages.

As above, we now recalculated the modest subducted slab length by considering the uncertainty of the duration of arc magmatism.

Line 268: Plate tectonics, yes, but not modern-style. Calling it 'modern-like' is confusing.

Thanks, we agree. So as not to cause debate, we have deleted the term “modern-like”. Also, we added the sentence “By the middle Paleoproterozoic (ca. 2.2 or 2.1 Ga), widespread high-pressure/low-temperature metamorphic rocks, large-scale collisional orogenesis, and amalgamation of the first widely-accepted supercontinent signify the latest timing of the establishment of the “modern-style” plate tectonic regime, as many defined, characterised by the occurrence of cold and deep subduction/collision and globally-interconnected network of narrow plate boundaries (e.g., Brown et al., 2020; Wan et al., 2020, Sci. Adv.; Pereira et al., 2021, EPSL).” to explain the change of plate tectonic styles from Neoproterozoic to early Paleoproterozoic in the discussion section.

Brown, M., Johnson, T. & Gardiner, N. J. Plate Tectonics and the Archean Earth. *Annu. Rev. Earth Pl. Sc.* **48**, 291-320 (2020).

Pereira, I. et al. Detrital rutile tracks the first appearance of subduction zone low T/P paired metamorphism in the Palaeoproterozoic. *Earth Planet. Sc. Lett.* **570**, 117069 (2021).

Wan, B. et al. Seismological evidence for the earliest global subduction network at 2 Ga ago. *Science Advances.* **6**, c5491 (2020).

Line 282: I think this is overestimated. How would a >1000 km oceanic plate not start cooling down and potentially generate a lower T/P gradient when subducted?

We agree. Based on our previous detailed study in the adjacent Dengfeng Complex (Huang et al., 2020 EPSL), there were intermediate T/P metamorphic rocks (~440 °C/GPa) in the accretionary complex unit and the P-T conditions overlap with the Neoproterozoic to present-day subduction geotherms (as marked in Fig. 7). For the Angou Complex, the inferred upper-plate TTG–arc volcanic unit in the Western and Central belts, the metamorphic rocks lack rutile and underwent higher-temperature metamorphism (as evidenced by the TTG anatexis and metamorphic zircons in sample 22RZ06b), whereas the MORB unit of the Eastern Belt preserves metabasaltic rocks containing rutile (dark brown, as shown below), high-Al amphibole, and Na-rich plagioclase, which suggests relatively low temperature and intermediate pressure (>1 GPa in general). Such metamorphic configuration is comparable to the paired metamorphism in the Dengfeng Complex as we proposed before. A more detailed metamorphic discussion is out of the scope of this manuscript, but will be presented in detail in our near future work.

Photomicrograph showing two dark brown rutile grains in a metabasalt (18AG09-1) from the MORB unit of the Eastern Belt of the Angou Complex.

Huang, B., Kusky, T.M., Johnson, T.E., Wilde, S.A., Wang, L., Polat, A. & Fu, D., Paired metamorphism in the Neoproterozoic: A record of accretionary-to-collisional orogenesis in the North China Craton. *Earth Planet. Sc. Lett.* **543**, 116355 (2020).

Line 324: mid not 'mod'.

Revised.

Line 352: The Huronian glaciation extends to 2.2/2.1 Ga.

Revised.

Line 366: This section of the paper can be improved. As it stands, it does not give enough detail for a proper assessment of the quality of data. Please, provide the analysed standards for U-Pb or Lu-Hf analyses.

As above, we have improved the description of zircon U-Pb and Lu-Hf analyses, and provide detailed information on the standards in **Tables S2–S3**.

Supplementary material -

Fig. S9: Caption says the weighted age is provided, but I can't see it anywhere. Could you also add the U-Pb and Lu-Hf spot location of analysed grains?

A careless error on our part, the concordia diagram was inadvertently omitted here. We have now added the zircon concordia diagram in Fig. S11 and updated the spot locations in Fig. S10.

=====

Reviewer #3:

To decipher the Archean plate tectonics activated prior to the present-day North China Craton, the authors divided the Angou Complex, southern Paleoproterozoic Trans-North China Orogen into three litho-structural belts: (1) the Eastern Belt, comprising metabasalt and fine-grained metasedimentary sequences, being regarded to be formed in the Archean spreading oceanic crust; (2) the Central Belt, composed of metamorphosed mafic to felsic volcanic rocks and sedimentary successions, ascribed to island arc basalt and related rocks; and (3) the Western Belt, consisting mainly of TTG gneiss and minor metabasite with the later being ascribed to MORB-like basalt. Age data indicate that tectonic juxtaposition of these three different belts occurred at or before 2.50 Ga, during arc–continent collisional orogenesis. The authors suggest that the Eastern Belt of the Angou Complex is one of the oldest passive margin sequences, whereas the Central and Western belts represent a SSZ arc/forearc setting. Combined with literature data, the authors rebuilt the late Archean plate tectonics of the “Proto-North China Ocean”. The results and inference are quite interesting. However, some questions should be considered and they are listed as follows.

We appreciate reviewer #3 very much for the careful, positive and helpful reviews. All comments/suggestions are insightful and constructive. We have revised the manuscript accordingly and gave a detailed reply to the comments as detailed below.

(a) Although the authors derived that the “Eastern Belt” of the Angou Complex formed at a spreading ridge outboard of a passive margin, but how could the BIF be formed at such tectonic setting? BIF is repeatedly certified to be formed at continental margin or shallow sea or lake in the Archean or Paleoproterozoic. BIF also appears in the “Central Belt”.

Good comment. Iron formation (IF) can form either on the deep-sea (generally >200 m) ocean floor near the mid-ocean ridge or in submarine island arc-related basins (forearc or

backarc) associated with volcanism and hydrothermal events (known as the Algoma-type BIF), or the shallow-sea continental shelf far from volcanic centers (known as the Superior-type IF) (e.g., Gross, 1980; Bekker et al., 2010; Konhauser et al., 2017). For many Archean BIFs, the Algoma-type IF is dominant; whereas in the Paleoproterozoic, the Superior-type IFs become dominant (typically granular iron formation, GIF) (Bekker et al., 2010, 2014). These two types of BIFs have significant differences both in terms of their host rocks (submarine volcanic rocks vs. siliciclastic sedimentary rocks, e.g., shale), scales (smaller and thinner vs. large and economic iron deposits, like those formed in West Australia), geochemical compositions (detritus-free and less continental flux vs. detritus-bearing with more continental flux) and redox and depositional environments.

In our previous work on the BIF in the Dengfeng Complex, adjacent to the Angou Complex, we reported geochemical and Sm-Nd isotopic evidence to show that the majority of BIFs belong to the Algoma-type, and were deposited near the volcanic and hydrothermal centers on the ocean-floor (mid-ocean ridge and forearc), with minor detritus-rich, BIF layers deposited closer to the continental margin (Huang et al., 2019, *Precambrian Res.*). For the BIFs in the Angou Complex, this model can explain their occurrence, which is closely associated with the MORB-chert ocean plate stratigraphy (the Eastern Belt) and arc-forearc volcanic and sedimentary rocks (chert-shale, the Central Belt), indicating a relatively deep-water environment. The details are beyond the scope of this paper, but we consider that our model is reasonable and applicable.

- Gross, G. A. A classification of iron formations based on depositional environments. *The Canadian Mineralogist*. **18**, 215-222 (1980).
- Bekker, A., Slack, J.F., Planavsky, N., Kraepz, B., Hofmann, A., Konhauser, K.O., Rouxel, O.J., Iron Formation: The Sedimentary Product of a Complex Interplay among Mantle, Tectonic, Oceanic, and Biospheric Processes. *Econ. Geol.* **105**, 467-508 (2010).
- Bekker, A., Planavsky, N.J., Kraepz, B., Rasmussen, B., Hofmann, A., Slack, J.F., Rouxel, O.J. & Konhauser, K.O., Iron Formations: Their Origins and Implications for Ancient Seawater Chemistry. *Treatise on Geochemistry 2nd Edition*: Elsevier; 2014. pp. 562-628.
- Konhauser, K. O., Planavsky, N.J., Hardisty, D.S., Robbins, L.J., Warchola, T.J., Haugaard, R., Lalonde, S.V., Partin, C.A., Oonk, P.B.H., Tsikos, H., Lyons, T.W., Bekker, A. & Johnson, C.M., Iron formations: A global record of Neoproterozoic to Palaeoproterozoic environmental history. *Earth-Sci. Rev.* **172**, 140-177 (2017).
- Huang, B., Kusky, T.M., Wang, L., Deng, H., Wang, J., Fu, D., Peng, H. & Ning, W., Age and genesis of the Neoproterozoic Algoma-type banded iron formations from the Dengfeng greenstone belt, southern North China Craton: Geochronological, geochemical and Sm–Nd isotopic constraints. *Precambrian Res.* **333**, 105437 (2019).

(b) The “Western Belt” of the Angou Complex consists mainly of TTG gneiss and minor amphibolite, which signify continental rather than oceanic origin.

As shown in our tectonic cartoon (Fig. 6c), the exposed TTG gneisses in the Western Belt of the Angou Complex represent eroded deep intra-oceanic arc roots that are dominated by highly-deformed felsic plutons (i.e., TTG) and subordinate mafic rocks. The adakite-like TTG magma intruded into the central layer of the intra-oceanic arc nuclei due to slab melting in the Neoproterozoic.

The middle and upper levels of the arc are composed of SSZ-type basaltic to rhyolitic volcanic suites and submarine sediments including local chert-shale-BIF in the Central Belt of the Angou Complex, indicating a relatively deep submarine setting rather than a continental setting. In the modern Andes or Cordillera continental margins, the rocks are dominated by plutonic, volcanic, and continental facies sedimentary rocks, rather than deep-water sediments. In intra-oceanic arcs, the upper levels are dominated by similar volcanic and sedimentary successions, and its uneroded and unexposed deep roots are interpreted to be mafic to felsic plutonic rocks, comparable with our observations in the Angou Complex.

In the Neoproterozoic, slab melting is the major mechanism for generating adakite-like rocks, whereas arc magmatism in the Phanerozoic is dominated by the melting of hydrous mantle/lower crust and its subsequent crystal fractionation, thus the proportions and compositions of felsic rocks may be different. Neoproterozoic intra-oceanic arcs were dominated by TTG and overlying SSZ-type basaltic to rhyolitic volcanic suites and submarine sediments, as observed in the Angou Complex. Our tectonic inference can well match the observed geological phenomena in the Angou Complex. Thus, we think that our interpretation is reasonable and convincing.

(c) All the rocks were metamorphosed at different extent. Therefore, the authors are suggested to specify what geochemical data (e.g., immobile elements) can be used to discriminate the tectonic settings of these rocks, in order to yield unambiguous results.

Good comment, we have added additional discussion on element mobility in the Supplementary Text. We mainly used the immobile elements in the discussion.

(d) About thermodynamic modeling. Because the authors used averaged MORB (MORB-AV) and IAB (IBA-AV) compositions as starting materials, the results should be ubiquitous to all such rocks. Please specify this. Furthermore, it seems there are similar modeling work already reported in the literature. So, please describe what are the differences between the modeling results of this manuscript and others'?

Thanks for the insightful comment. Due to limitations on the length of the manuscript, the details of this work were not fully explained in the previous version. We used different potential source rocks to test popular geodynamic models. Compared with previous modelling work on high-pressure TTG origins, our modelling results support the slab-melting model, rather than many recently-popular models such as partial melting of subducted arc basalt, of the base of the oceanic plateau, of intra-plate basalt, or pure fractional crystallization of mantle-derived magma. We think our modelling logic flow is reasonable, and the consequent results on subduction zone geotherms are novel and robust as explained below.

With the development and improvement of the internally-consistent thermodynamic dataset (Holland and Powell, 2011) and mineral activity models (e.g., White et al., 2014; Green et al., 2016; Holland et al., 2020), thermodynamic and trace element modelling can be applied to model various partial melting processes using different source rocks (starting materials) (either metapelitic or metamafic rocks) and P-T-XH₂O conditions, with the growing literature published these years in high-level journals that shed light on the petrogenetic and geodynamic complexity of TTG (e.g., Nagel et al., 2012; Palin et al., 2016 that mainly focused on the major elements; Johnson et al., 2017; Ge et al., 2018; Johnson et al., 2019; Sun et al., 2021) and other granitic rocks (e.g., Collins et al., 2020; Pourteau et al., 2021). The compositions of starting materials exert a major control on the modelling results. Available modelling uses two methods:

(i) method 1: using the average compositions of global or regional rock datasets, e.g., the average Archean basalts (e.g., Ge et al., 2018; Palin et al., 2016), MORB (Palin et al., 2016), and average regional basalts from a specific period without distinguishing different rock types (e.g., 2.9 Ga average basalt in the eastern North China craton, Sun et al., 2019). This method using global or regional average compositions as starting materials is generally used to decipher the general trends of partial melting processes of mafic rocks,

but it is generally hard to constrain the precise conditions and processes of partial melting in specific cases as it didn't consider the heterogeneity of source rocks in different regions and time.

(ii) method 2: compositions of local potential source rocks that are associated with target TTG rocks (one typical sample is Nagel et al., 2012, Collins et al., 2020; or an average composition of certain-type rocks, Johnson et al., 2017, 2019). The second method using the local composition of potential source rocks is much closer to natural situations and widely used in constraining the P-T conditions of partial melting in many special cases (e.g., Nagel et al., 2012; Johnson et al., 2017, 2019; Collins et al., 2019; Pourteau et al., 2020). For example, using the MORB-like composition as the starting material, we can model the partial melting of the subducting slab; using the IAB-like composition, we can model the partial melting of thickened arc crust or subducted arc crust; using OIB/oceanic plateau basalt-like compositions, we can model the partial melting of subducting oceanic plateau, or melting at the base of an oceanic plateau. This method can directly link the melt compositions with source rock compositions and P-T conditions, thus providing greater constraints on the geodynamic setting. Note that the MORB- and IAB-like rocks in different regions and times may also be slightly different in their major and trace element compositions, thus it's more convincing to use the local rocks that are associated with the TTG.

In our work, we use the second method — average compositions of different rock types in the study area as the starting compositions to test potential geodynamic models: 1) partial melting of the subducting slab (modelled P-T results should be high-P/intermediate-T) using MORB-type rock composition, and 2) partial melting of the base of the arc basalt (intermediate-P and high-T), or subducting arc basalt (high-P/intermediate-T) using IAB-type rock composition. Our modelled results preclude the possibility of partial melting of IAB-type rocks in generating TTG rocks in the Angou and Dengfeng complexes; and show that the partial melting of MORB-type rocks (subducting slab) at high-P and intermediate-T conditions can best match the TTG rocks in the study area.

Therefore, we consider our logic reasonable. Also, our modelling result supports the classic slab melting model, rather than recently popular non-subduction models or melting of subducting arc or thickened arc. In addition, our modelling provides novel constraints on the thermal states of Neoproterozoic subduction zones, at least in the southern North China

Craton.

- Nagel, T. J., Hoffmann, J. E. & Münker, C. Generation of Eoarchean tonalite-trondhjemite-granodiorite series from thickened mafic arc crust. *Geology*. **40**, 375-378 (2012).
- Palin, R. M., White, R. W. & Green, E. C. R. Partial melting of metabasic rocks and the generation of tonalitic-trondhjemitic-granodioritic (TTG) crust in the Archaean: Constraints from phase equilibrium modelling. *Precambrian Res.* **287**, 73-90 (2016).
- Johnson, T. E., Brown, M., Gardiner, N. J., Kirkland, C. L. & Smithies, R. H. Earth's first stable continents did not form by subduction. *Nature*. **543**, 239-242 (2017).
- Johnson, T. E. et al. An impact melt origin for Earth's oldest known evolved rocks. *Nat. Geosci.* **11**, 795-799 (2018).
- Ge, R., Zhu, W., Wilde, S. A. & Wu, H. Remnants of Eoarchean continental crust derived from a subducted proto-arc. *Science Advances*. **4**, o3159 (2018).
- Pourteau, A., Doucet, L.S., Blereau, E.R., Volante, S., Johnson, T.E., Collins, W.J., Li, Z., Champion, D.C., TTG generation by fluid-fluxed crustal melting: Direct evidence from the Proterozoic Georgetown Inlier, NE Australia. *Earth Planet. Sc. Lett.* **550**, 116548 (2020).
- Collins, W. J., Murphy, J. B., Johnson, T. E. & Huang, H. Critical role of water in the formation of continental crust. *Nat. Geosci.* **13**, 331-338 (2020).
- Sun, G., Liu, S., Cawood, P.A., Tang, M., van Hunen, J., Gao, L., Hu, Y., Hu, F., Thermal state and evolving geodynamic regimes of the Meso- to Neoproterozoic North China Craton. *Nat. Commun.* **12**, (2021).

Specific comments:

Abstract: The study area should be given clearly.

We added the exact study area in the abstract.

Lines 40 and 42. "Earth" can be revised to be "Earth's".

Revised.

Lines 66, 258 and 259. Please check and revise the term "~1600 km-long.....". In fact, the Trans-North China Orogen is no more than 500 km long.

Thanks for pointing out this. We checked this carefully, and confirmed the Neoproterozoic Central Orogenic Belt reaches ~1600-km long; the N-S-trending length of the Paleoproterozoic Trans-North China Orogen is more than 1200 km and the E-W width is <500 km.

The central orogen of the North China Craton has been named "Trans-North China Orogen" (e.g., Zhao et al., 2001, 2005, 2012) or "Central Orogenic Belt" (e.g., Kusky et al., 2001; Kusky and Li, 2003; Kusky et al., 2016). The former is defined as a Paleoproterozoic (~1.85

Ga) continent-continent collisional orogen between the Eastern and Western blocks based mainly on their difference in the Neoproterozoic (dominantly counter-clockwise P-T paths) and Paleoproterozoic (clockwise P-T paths) metamorphic styles; whereas the latter emphasized Neoproterozoic (~2.5 Ga) arc-continent collisional orogenesis based on Neoproterozoic subduction–accretion–collision related mélanges, arc magmatism, and suture zones. The exact boundaries between these two definitions are not precise in some regions due to the limitation of outcrops, and they have minor differences, but the scale is comparable. In this work, we mainly focus on the Neoproterozoic subduction and collisional orogenesis, thus we use the term of the Neoproterozoic “Central Orogenic Belt” for this tectonic unit. After a careful check of the distribution of the central orogen, we measured the NE-SW-trending length of the Neoproterozoic Central Orogenic Belt (from North Liaoning to South Henan) on Google Earth and confirmed that the orogen reaches more than ~1600 km long.

Zhao, G., Wilde, S. A., Cawood, P. A. & Sun, M. Archean blocks and their boundaries in the North China Craton: lithological, geochemical, structural and P–T path constraints and tectonic evolution.

Precambrian Res. **107**, 45-73 (2001).

Zhao, G., Sun, M., Wilde, S. A. & Li, S. Late Archean to Paleoproterozoic evolution of the North China Craton: key issues revisited. *Precambrian Res.* **136**, 177-202 (2005).

Zhao, G., Cawood, P.A., Li, S., Wilde, S.A., Sun, M., Zhang, J., He, Y., Yin, C., Amalgamation of the North China Craton: Key issues and discussion. *Precambrian Res.* **222-223**, 55-76 (2012).

Kusky, T. M. & Li, J. Paleoproterozoic tectonic evolution of the North China Craton. *J. Asian Earth Sci.* **22**, 383-397 (2003).

Kusky, T.M., Polat, A., Windley, B.F., Burke, K.C., Dewey, J.F., Kidd, W.S.F., Maruyama, S. et al. Insights into the tectonic evolution of the North China Craton through comparative tectonic analysis: A record of outward growth of Precambrian continents. *Earth-Sci. Rev.* **162**, 387-432 (2016).

Line 82. “2.50 Ga” can be corrected as “~2.45 Ga”.

Corrected.

Lines 125-127. To decide the maximum depositional age of a sedimentary sequence, only the youngest age of detrital zircons can be used. The authors should explain why 2533±13 Ma was estimated to be the maximum depositional age.

Thanks for this comment. The popular methods to estimate the maximum depositional ages (MDA) using detrital zircon U-Pb dating include 1) the youngest single age; 2) the weighted mean age (or peak age) of the youngest group of ages (e.g., 3–5); and 3) the youngest peak age (Dickinson and Gehrels, 2009; Gehrels, 2014). After precluding contamination, methods 1 or 2 can be used for many Phanerozoic strata. This estimate would, however, be

difficult for Precambrian sedimentary rocks, because of measurement uncertainties, and loss of Pb, so other statistical methods need to be used.

To better explain this, we can assume a sandstone with its all detritus coming from a ~100-Ma granite (Gehrels, 2013). We obtained a set of zircon data, and none of the analyses is compromised by Pb loss or inheritance. In an ideal world, all analyses would be exactly 100.0 Ma. Unfortunately, none of the analyses will yield an age of exactly 100.0 Ma because of measurement uncertainty (generally 2% by LA-ICPMS) (Gehrels, 2013). The resulting measured ages might scatter from <98 to >102 Ma, following the Gaussian distribution (normal distribution). If we calculate the MDA using the Youngest Single Age, the youngest age from the set would be 97.6 ± 0.8 Ma in this case. As shown in this example, the youngest single grain will always be younger than the true age, and should never be used as the MDA when it is part of a cluster of overlapping ages (see figure below; and more detailed explanation from Gehrels, 2013).

An example showing the effect of measurement uncertainty, and comparisons between two main calculation methods of maximum depositional age (MDA) (From Gehrels, 2013)

For Precambrian zircons, the $\sim \pm 2$ % analysis error of LA-ICPMS zircon dating would cause much significant influence on the resulting measured ages. For example, if all detrital

zircons of sandstone have ages of 2500 Ma, the youngest single zircon age would be ~2450 Ma, which is younger than their true age (~2500 Ma) by 50 Ma. After precluding systematic equipment errors, only the weighted mean age (or peak age) of all analysed zircon ages can yield the “closest” age of ~2500 Ma. Thus, for the early Precambrian sedimentary rocks without significant Pb loss, the weighted mean age of the youngest peak would be one of the most appropriate and robust estimates of MDA.

Thus, in our case, the sedimentary was deposited in the near forearc, it's reasonable and better to use the youngest peak age (here just one peak) of ~2533 Ma as the MDA.

Dickinson, W. R. & Gehrels, G. E. Use of U–Pb ages of detrital zircons to infer maximum depositional ages of strata: A test against a Colorado Plateau Mesozoic database. *Earth Planet. Sc. Lett.* **288**, 115-125 (2009).

Gehrels, G. Detrital zircon U-Pb geochronology applied to tectonics. *Annu. Rev. Earth Pl. Sc.* **42**, 127-149 (2014).

Gehrels, 2013, MDA Tutorial: Determining Max Depo Ages,
<https://sites.google.com/laserchron.org/arizonalaserchroncenter/home>

Line 167. “is” is suggested to corrected as “are”.

Corrected.

Line 165. About the “Thermodynamic and trace element modelling” section. The author may cite direct petrological experiments if there are such data, rather than just on numerical modeling results.

Thanks for this suggestion. The internally consistent thermodynamic data (Holland and Powell, 2011) and mineral–melt partition coefficients (Bedard, 2006) used here are calibrated from the results of petrological experiments. Using these petrological experiment-constrained datasets, we can reliably conduct forward thermodynamic and trace element modelling on partial melting processes of different source rocks at a given P–T space. This method has been demonstrated to be valid. We also cited a classical petrological experiment reference (Foley et al., 2002).

Foley, S., Tiepolo, M. & Vannucci, R. Growth of early continental crust controlled by melting of amphibolite in subduction zones. *Nature.* **417**, 837-840 (2002).

Line 317. About the “Implications for the Neoproterozoic-Proterozoic surficial environments”. This section does not match the topic of this manuscript and can be deleted.

Thanks for this suggestion. In this section, we explored the possible link between global plate tectonic processes with the surficial environments during the Neoproterozoic-early Paleoproterozoic transformative period of Earth's history. We think this section is related to the core of the paper that concentrates on the documentation of plate boundary assemblages, and provides constraints on the plate tectonic style in the late Neoproterozoic, which may have had a profound influence on surficial environments at that time. Significantly, we have discussed the roles of such plate boundary processes, such as mid-ocean ridge, continental margin, subduction zones, and arc-continent collision in shaping the continent, ocean, atmosphere and geosphere. All discussion and conclusions are based on the integration of reliable datasets (e.g., metamorphism, seawater Sr, passive margin, and atmosphere, etc) and models (e.g., lateral crustal growth, erosion of mountain belts, silicate weathering, GOE, etc) (see references cited), and our observations/findings such as the seafloor spreading, intra-oceanic subduction, arc-continent collision, and foreland sequences. The section could be of interest to the geology community to better link the plate tectonic processes with its environmental response and the establishment of our habitable planet in such one of the most transformative periods of Earth's history (as we noted in the Introduction section). Thus, we hope we can keep this section in the paper, which may attract interest from the general geology readers as promoted by the journal.

Line 324. "mid-ocean" should be corrected as "mid-ocean".

Corrected.

The reference list: The cited journals are in either full names or abbreviations, so the style should be unified.

Thanks, we carefully checked and revised citations of references following the style as the journal required.

Line 413. Authors are lost.

We carefully checked this; no special authors are listed.

<https://doi.org/10.1038/s41467-020-19930-3>

Line 499. "Lithosphere-US." Should be corrected as "Lithosphere".

Done.

Line 547. “advances” should be corrected as “Advances”.

Done.

Figure 1 is unnecessary but can be deposited as an electronic appendix.

Fair point, we agree. We have now moved this figure to the supporting information.

Figure 7a. What are the pink regions?

Thanks for pointing out this. The pink regions represent the TTG-dominated felsic arc crust. We added the explanation in the legend.

Figure 8. Routine description of thermal gradient is in °C/kbar or °C/km rather than in °C/GPa.

The transition from °C/kbar or °C/GPa to °C/km needs to know the conversion relation between pressure (kbar or GPa, lithostatic pressure plus possible tectonic overpressure) and depth (km), which involves uncertainty about rock density and tectonic overpressure. Using °C/kbar or °C/GPa is a more direct and objective way. Given that °C/GPa has been also popularly used in recent metamorphic studies, and that our calculated pressure of partial melting is ~1.6–1.8 GPa, we have kept °C/GPa and added a comment (1 GPa = 10 kbar) in the figure caption, so readers can easily understand.

REVIEWERS' COMMENTS

Reviewer #1 (Remarks to the Author):

I have now finished my review of the aforementioned Nature Communications Manuscript which I provide below.

MS DESCRIPTION & SYNOPSIS On the basis of whole rock major and trace element, zircon U-Pb and Lu-Hf isotopes, phase equilibrium modeling and trace element modeling, the authors provide a model stipulating the operation of plate tectonics circa 2.5 Ga.

PRESENTATION & SCIENTIFIC INTERPRETATION As I wrote in my original review, which is again true for this version, the text is very well-written with few syntax or spelling mistakes. The ms will be of broad interest to those working on the origin and timing of plate tectonics on Earth.

A couple of general comments: (1) I completely agree with Reviewer 2 (Moreira) that "this is a robust work with detailed field documentation, which is rare in the recent literature that reached similar conclusions; and deserves publication in Nature Communications"; and (2) that the authors did an exemplary job at addressing the comments of all three reviewer of the original manuscript. I still have a few questions and comments which can be seen on the annotated 2 PDF version of the manuscript that I provide and I suspect that these are more due to my lack of detailed knowledge than those of the authors. I highly recommend publication in Nature Communications.

Reviewer #2 (Remarks to the Author):

The paper by Huang and others offers compelling evidence for the operation of plate tectonics in the Neoproterozoic. I think this contribution is timely and much needed for the current debate concerning global plate tectonics operation in the early Earth. Most of my comments on the previous manuscript version have been carefully addressed, and the remaining points were clarified, explained with further detail or plausibly rebutted. For the U-Pb analyses, I wholeheartedly recommend plotting the data at 2 sigma confidence level instead of 1 sigma (as reported in supplementary material). I am looking forward to seeing this contribution published.

Best wishes,

Hugo Moreira – September 2022

Reviewer #3 (Remarks to the Author):

It seems that all the problems proposed by the referees have been answered by the authors.

Response to Reviewers

POINT-BY-POINT RESPONSE TO REVIEWER COMMENTS (reply in blue)

Reviewer #1

Review of NCOMMS-22-07342A:

Dear Editor,

I have now finished my review of the aforementioned *Nature Communications* Manuscript which I provide below.

MS DESCRIPTION & SYNOPSIS

On the basis of whole rock major and trace element, zircon U-Pb and Lu-Hf isotopes, phase equilibrium modeling and trace element modeling, the authors provide a model stipulating the operation of plate tectonics circa 2.5 Ga.

PRESENTATION & SCIENTIFIC INTERPRETATION

As I wrote in my original review, which is again true for this version, the text is very well-written with few syntax or spelling mistakes. The ms will be of broad interest to those working on the origin and timing of plate tectonics on Earth.

A couple of general comments: (1) I completely agree with Reviewer 2 (Moreria) that “this is a robust work with detailed field documentation, which is rare in the recent literature that reached similar conclusions; and deserves publication in *Nature Communications*”; and (2) that the authors did an exemplary job at addressing the comments of all three reviewers of the original manuscript. I still have a few questions and comments which can be seen on the annotated PDF version of the manuscript that I provide and I suspect that these are more due to my lack of detailed knowledge than those of the authors. I highly recommend publication in *Nature Communications*.

We thank Prof. Whattam for the positive recommendation and constructive comments in the past two rounds of reviews. The new comments have been revised and concerns have been addressed accordingly.

The annotated comments in the PDF were listed as follows:

Line 102: “Methods”, I am a little confused as to why Methods does not precede Results. Is this Nature Communications protocol? At least indicate here that these are LA-ICP-MS ages and dating is not described in Methods.

Yes, the “Methods” section should be located in a separate section after “Discussion” as required by the journal. The results are presented in supplementary tables as noted in the “Methods”.

Lines 116-117: “and an associated continental margin”, I don’t understand this.

We revised the sentence as “and a continental margin”.

Line 142: Maybe also indicate La/Yb.

Yes, all samples also have high La/Yb ratios. We now added the La/Yb_{cn} ratios here.

Line 158: OK, but some may argue elevated Ba represents hydrothermal alteration; hence my comment on first manuscript pertaining to filtering-i.e., removing high LOI samples.

We have removed samples with high LOI (> 5 wt%), and discussed the elemental mobility.

Lines 159-160: not sure if this can/should be stated.

We consider that this can be kept here. Such geochemical characteristics are consistent with these of forearc basalts (FAB), with a slight contribution of subduction-related fluids (Fig. 3). Later in the discussion, we also have a further discussion that the assemblages from MORB-like basalt (FAB or MORB), IAB, volcanic rocks and high-Mg diorite to adakitic rocks are consistent with the magmatic progression from subduction initiation to arc maturation. The MORB-like metabasalts are the oldest and thus best explained as the FAB.

Line 166: What about La/Yb?

They also have high La/Yb ratios, consistent with adakitic features. We added the La/Yb_{cn} ratios here.

Lines 177 and 189: ok, but models of adakite formation are diverse.

Thanks for this good comment. We revised the relevant sentences here to make it clearer.

Although there are different models for explaining the generation of adakitic/TTG magmas, the popular petrogenetic models include partial melting, fractional crystallization (FC), and magma mixing. The FC process may play an important role in magmatic evolution and compositional diversity, but most adakitic suites lack continuous geochemical evolutionary trend from basaltic to rhyolitic rocks, suggesting that they are not the mechanism in generating the majority of primitive adakitic/TTG magmas. Given the general lack of evidence for the mixing of mantle- and crust-derived magmas (e.g., microgranular mafic enclaves in adakitic rocks) in many magmatic suites, the magma mixing model may also have problems in explaining the adakitic/TTG origins.

Therefore, the partial melting model has been regarded as playing a major role in the formation of primitive adakitic or TTG magmas. There is a general consensus that partial melting to generate adakitic and TTG magmas requires a hydrous mafic source, although with diverse geodynamic models (e.g., partial melting of subducting slab, island arc, or oceanic plateau; partial melting at the base of oceanic plateau or thickened lower crust, etc). Based on the geochemical analysis and thermodynamic and trace element modeling, here we propose that the partial melting of subducting slab and subsequent FC process had contributed to the generation and evolution of the more primitive tonalitic (Dengfeng) and trondhjemitic (Angou) TTG suites, respectively.

Line 219: change “the” into “a”.

Done.

Line 244: what about uncertainties?

Fig. S13 shows the uncertainties of these rocks.

Line 279: add reference 41.

Done.

Line 309: I don't understand “bimodal thermobaric ratios (T/P)”.

This means bimodal temperature/pressure ratios in metamorphic rocks, which is generally expressed as “paired metamorphism”, i.e., coexisting high-P-low-T (subducting zone) and high-T-low-P (arc/backarc) metamorphism.

Line 314: “juxtaposed”, how? see comment on Fig. 6: Still don't understand why MOR assemblage

is presented, and not entirely subducted.

As we replied in detail in the first version, the MOR-type OPS (ocean plate stratigraphy) can be accreted to the forearc, forming an accretionary complex (relatively coherent and repeated OPS unit and chaotic block-in-matrix mélange unit). This is common in many accretionary plate margins, like Japan and Alaska (see reviews by Kusky et al., 2013, 2020 that have been listed in the last response).

Kusky, T. M., Windley, B.F., Safonova, I., Wakita, K., Wakabayashi, J., Polat, A., Santosh, M.,
Recognition of ocean plate stratigraphy in accretionary orogens through Earth history: A record of
3.8 billion years of sea floor spreading, subduction, and accretion. *Gondwana Res.* **24**, 501-547
(2013).

Kusky, T. et al. Mélanges through time: Life cycle of the world's largest Archean mélange compared
with Mesozoic and Paleozoic subduction-accretion-collision mélanges. *Earth-Sci. Rev.* **209**, 103303
(2020).

Line 335: Amphibolite and eclogite are simply formed due to subduction of basaltic crust; what refs. support MORB slab melting = amphibolite-eclogite?

Thanks, as shown in Fig 8, the MORB slab can be partially melted, with amphibolite, granulite and eclogite facies residues in the warmer subduction zones; The cold subduction zones would be dominated by the slab dehydration in the shallow part. The melting of mafic rocks (including MORB) to generate TTG/adakitic magmas have been supported by petrological experiments as reviewed by Moyen and Martin (2012).

Moyen, J. & Martin, H. Forty years of TTG research. *Lithos.* **148**, 312-336 (2012).

Line 336: "off-scraped" typo.

Thanks, corrected as "offscraped".

Line 339: "impetus" what does this mean in this context?

This means that slab residues have negative buoyancy to maintain the oceanic spreading, slab subduction and melting.

Line 345: "modern average MORB mantle" Ref.

Done.

=====

Reviewer #2 (Hugo Moreira):

The paper by Huang and others offers compelling evidence for the operation of plate tectonics in the Neoproterozoic. I think this contribution is timely and much needed for the current debate concerning global plate tectonics operation in the early Earth. Most of my comments on the previous manuscript version have been carefully addressed, and the remaining points were clarified, explained with further detail or plausibly rebutted. For the U-Pb analyses, I wholeheartedly recommend plotting the data at 2 sigma confidence level instead of 1 sigma (as reported in supplementary material). I am looking forward to seeing this contribution published.

Best wishes,

Hugo Moreira – September 2022

We sincerely appreciate Dr Hugo Moreira for the positive recommendation and constructive comments raised in the past two rounds of reviews. For zircon U-Pb diagrams, we have plotted the data at 2 sigma confidence in previous version, but shown data in 1 sigma. Here we revised the relevant ages to 2 sigma error in the revised manuscript.

=====

Reviewer #3:

It seems that all the problems proposed by the referees have been answered by the authors.

We appreciate reviewer #3 very much for the positive recommendation and helpful reviews.